# Gradient Regularization Mitigates Reward Hacking
# in Reinforcement Learning from Human Feedback and Verifiable Rewards

**Johannes Ackermann** [1 2]  **Michael Noukhovitch** [3 4]  **Takashi Ishida** [2 1]  **Masashi Sugiyama** [2 1]

## Abstract

Reinforcement Learning from Human Feedback (RLHF) or Verifiable Rewards (RLVR) are two key steps in the post-training of modern Language Models (LMs). A common problem is reward hacking, where the policy may exploit inaccuracies of the reward and learn an unintended behavior. Most previous works address this by limiting the policy update with a Kullback-Leibler (KL) penalty towards a reference model. We propose a different framing: Train the LM in a way that biases policy updates towards regions in which the reward is more accurate. First, we derive a theoretical connection between the accuracy of a reward model and the flatness of an optimum at convergence. Gradient regularization (GR) can then be used to bias training to flatter regions and thereby maintain reward model accuracy. We confirm these results by showing that the gradient norm and reward accuracy are empirically correlated in RLHF. We then empirically show that Reference Resets of the KL penalty find flatter regions with a higher reward accuracy. We further improve on this by proposing to use explicit GR with an efficient finite-difference estimate. Empirically, GR performs better than a KL penalty across a diverse set of RL experiments with LMs. GR achieves a higher GPT-judged win-rate in RLHF, avoids overly focusing on the format in rule-based math rewards, and prevents hacking the judge in LLM-as-a-Judge math tasks.

## 1. Introduction

Reinforcement Learning (RL) has become a key part of the post-training of language models (LMs) (Stiennon et al.,

2020; Shao et al., 2024). In the case of RL from Human Feedback (RLHF) (Christiano et al., 2017), we use RL to align the behavior of LMs with human preferences, which we cannot easily represent with a rule-based reward. In the case of RL from Verifiable Feedback (RLVR) (Havrilla et al., 2024; Lambert et al., 2024), RL is used to improve the performance on tasks with verifiable rewards, such as mathematical reasoning or agentic tasks. In RLHF, we use pairwise comparison data to train a reward model (RM), which then provides the reward estimates for policy updates. In RLVR, we use a verifier such as a rule-based reward or another Large Language Model (LLM) to check if the model output matches the expected answer. In both cases there is a desired behavior, corresponding to a desired true reward, which we try to approximate with the trained RM, rule-based reward, or LLM-as-a-judge reward. We collectively refer to them as proxy rewards (PRs). A key challenge of RL post-training is: How can we ensure that when updating our policy with the PR, we actually improve the true reward, i.e., how can we ensure that our PR stays accurate as the policy changes throughout training? One solution is to iteratively update the PR during training with new data from the current policy (Christiano et al., 2017). As this can be costly, another option is to use a Kullback-Leibler (KL) penalty to ensure the policy stays close to the initial model (Stiennon et al., 2020). In practice, the KL penalty slows down training and may not even improve performance (Gao et al., 2023), leading to recent papers abandoning it for tasks with rule-based rewards (GLM-4.5 Team et al., 2025; Olmo Team et al., 2025). However, this risks reward hacking with reward models and LLM-as-a-judge.

We thus aim to modify the policy update such that the policy not only maximizes the PR, but also maximizes the PR accuracy, without constraining it to stay close to the initial policy. We argue that reward hacking often corresponds to sharp optima of the PR and propose gradient regularization (GR) as a solution. We illustrate an overview in Figure 1.

In Section 3, we formalize this notion and show a theoretical connection between the flatness of an optimum and the PR accuracy at this optimum, as measured by the Bradley-Terry (BT; Bradley & Terry, 1952) loss. In Section 4, we show that this theoretical connection can be utilized to improve

[1]The University of Tokyo [2]RIKEN AIP [3]Mila [4]Université de Montréal. Correspondence to: Johannes Ackermann <ackermann@ms.k.u-tokyo.ac.jp>.

*Proceedings of the 43rd International Conference on Machine Learning*, Seoul, South Korea. PMLR 306, 2026. Copyright 2026 by the author(s).

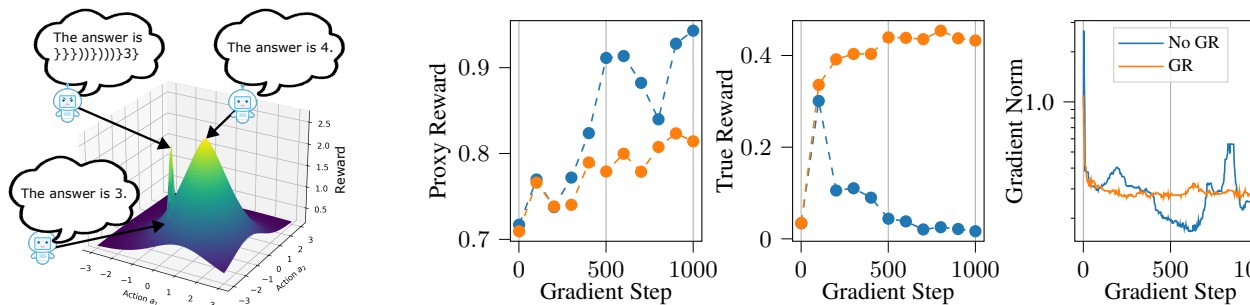

*Figure 1.* We argue that reward hacking often corresponds to exploiting sharp maxima in action space, as illustrated by the conceptual figure (left). For example, an LLM judge may be confused and assign a high reward to a wrong answer with specific formatting. In the LLM-as-a-Judge training run shown on the right, the increase in gradient norm coincides with reward hacking, resulting in true reward collapsing. By using gradient norm regularization, we can prevent this issue and obtain a better model, as seen by the improved true reward. The examples show Qwen2.5-0.5B models trained on GSM8K with a Qwen2.5-1.5B-Instruct judge with access to the true answer.

RLHF in practice. We leverage a recent method, Reference Resets (Liu et al., 2025a), and demonstrate that it implicitly regularizes the gradient, providing a novel interpretation of this method. We empirically show this enables better RLHF performance on the TL;DR task. In Section 5, we propose a novel application of explicit gradient regularization to RL post-training. We demonstrate that it outperforms baselines in RLHF tasks with reward models as well as rule-based and LLM-as-a-judge RLVR math tasks. By using GR, we are able to train proficient LLM-as-a-Judge and RLHF models *completely replacing the standard KL penalty.*

## 2. Background

### 2.1. Reinforcement Learning

As is common in the RL post-training literature (Shao et al., 2024; Ahmadian et al., 2024), we consider an episode consisting of a single state $s$, representing the prompt, and a single action $a$, representing the reply. We denote the state set by $\mathcal{S}$ and the action set by $\mathcal{A}$. As policy $\pi_\phi$ with parameters $\phi$, we consider an autoregressive LM with conditional probability $\pi_\phi(a|s) = \prod_t \pi_\phi(\mathfrak{a}_t|s, \mathfrak{a}_{<t})$, where $\mathfrak{a}_t$ is the $t$-th response token, $\mathfrak{a}_{<t}$ refers to the response tokens before $\mathfrak{a}_t$ and $a$ refers to the entire response. We assume that there is a true reward function $R^* : \mathcal{S} \times \mathcal{A} \to \mathbb{R}$ and our goal is to maximize the return of the policy $\pi_\phi$ on this reward $J^*(\phi) = \mathbb{E}_{s \sim P(s), a \sim \pi_\phi}[R^*(s, a)]$. However, we do not have access to $R^*$. Instead, we use a proxy reward (PR) $\widetilde{R}$, and train the policy to maximize its return $J(\phi, \theta) = \mathbb{E}_{s \sim P(s), a \sim \pi_\phi}[\widetilde{R}_\theta(s, a)]$. To optimize the return we use REINFORCE policy gradient updates (Williams, 1992)

$$\nabla_\phi J(\phi, \theta) =$$
$$\mathbb{E}_{s \sim P(s), a \sim \pi_\phi} \left[ (\widetilde{R}_\theta(s, a) - b(s)) \nabla_\phi \log \pi_\phi(a|s) \right] , \quad (1)$$

where $b(s)$ is a baseline. In this work, we use a variant of Group Relative Policy Optimization (GRPO) (Shao et al.,

2024) called GRPO Done Right (Dr.GRPO) (Liu et al., 2025b), which removes normalization terms from GRPO. As we do not do multiple updates per batch, Dr.GRPO simplifies to using REINFORCE with the baseline being the sample average over multiple actions drawn for the same state, i.e., $b(s) = \frac{1}{N} \sum_{i=1}^{N} \widetilde{R}(s, a^i)$, with $a^i \sim \pi_\phi(a|s)$. GRPO was originally presented with a KL penalty $D_{\mathrm{KL}}(\pi_\phi; \pi_{\phi^1})$, as commonly used in RLHF (Stiennon et al., 2020). This KL-penalty, weighted by a hyper-parameter $\beta$, is intended to keep the model close to the initial policy $\pi_{\phi^1}$, preventing reward hacking (Stiennon et al., 2020), but is sometimes omitted in more recent works (GLM-4.5 Team et al., 2025; Olmo Team et al., 2025).

### 2.2. Proxy Rewards

We consider three types of PRs: Trained reward models $R^\theta$, rule based rewards $R^{\mathrm{R}}$, and LLM-as-a-Judge rewards $R^{\mathrm{LM}}$.

**Reward Models** (RMs) are commonly used in RLHF (Christiano et al., 2017; Stiennon et al., 2020) to represent complex preferences which are hard to turn into rule-based rewards. They generally assume the Bradley-Terry (BT) model of preference (Bradley & Terry, 1952) where the probability $P$ of preferring one option $a_1$ over another option $a_0$ is the logistic function $\sigma(x) = 1/(1 + e^{-x})$ of the difference of the true reward for each action, i.e.,

$$P(a_1 > a_0 \mid s) = \sigma\left(R^*(s, a_1) - R^*(s, a_0)\right) . \quad (2)$$

Pairs of responses $(a_0, a_1)$ for each prompt $s$ are collected from an initial model $\pi_{\phi^1}$. Human annotators then choose their preferred (winning) reply $a_{\mathrm{w}}$ and not preferred (losing) reply $a_1$, creating a dataset $\mathcal{D}_{\mathrm{RM}} = \{(s^j, a_{\mathrm{w}}^j, a_1^j)\}_{j=1}^{N_{\mathrm{RM}}}$. We can then use the BT assumption to train an RM $R_\theta : \mathcal{S} \times \mathcal{A} \to \mathbb{R}$ with parameters $\theta \in \Theta$ by minimizing the cross-entropy loss based on the BT model:

$$\mathcal{L}_{\mathrm{BT}}(\theta, \phi) =$$
$$- \mathbb{E}_{(s, a_{\mathrm{w}}, a_1) \sim P_{\pi_\phi}} [\log \sigma(R_\theta(s, a_{\mathrm{w}}) - R_\theta(s, a_1))] , \quad (3)$$

where $P_{\pi_\phi} = P(s)\pi_\phi(a_0|s)\pi_\phi(a_1|s)P(a_1 > a_0|s)$ is the probability of the $(s, a_{\mathrm{w}}, a_{\mathrm{l}})$ triplet under the policy $\pi_\phi$. To train the RM we minimize $\mathcal{L}_{\mathrm{BT}}(\theta, \phi^1)$ and the expectation is replaced by the sample average from $\mathcal{D}_{\mathrm{RM}}$.

**Rule-Based Rewards** (Havrilla et al., 2024) are deterministic checks whether a final answer matches the ground truth. To encourage reasoning, and to allow a discrimination of a final answer against occurrences of the answer during reasoning, rule-based rewards typically require the answer to follow a specific format, for example using the LaTeX tag \boxed{} (Hendrycks et al., 2021) or the HTML tags (DeepSeek-AI, 2025) <think>...</think><answer>...</answer>. Thus, oftentimes one reward term $R^{\mathrm{F}}$ checks whether the format matches and another reward term $R^{\mathrm{C}}$ checks for correctness, to give the rule-based reward $R^{\mathrm{R}} = R^{\mathrm{C}} + R^{\mathrm{F}}$.

**LLM-as-a-Judge** (Zheng et al., 2023) prompts an LLM and uses its textual output to check whether an answer is correct. Frequently designing a rule-based reward can be challenging due to many possible accurate solutions, e.g., the correctness of a proof or many equivalent ways to write a math answer. Instead, prompting an LLM-as-a-Judge with a description of scoring criteria, the question, and the correct answer (if available) can allow us to capture solutions more robustly. LLM-as-a-Judge can also allow for more complicated rewards with a combination of objective (e.g., correct reply) and subjective (e.g., clear reasoning) criteria.

### 2.3. Gradient Regularization (GR)

Flat minima of a loss function $\mathcal{L}(\phi)$ are connected to better generalization in supervised learning (Hochreiter & Schmidhuber, 1997; Foret et al., 2021), i.e., a smaller difference between the population/test loss $\mathcal{L}(\phi) = \mathbb{E}_{(x,y)\sim P(x,y)}[\ell(f_\phi(x), y)]$ and its empirical approximation on a finite training dataset $\mathcal{D}$, consisting of i.i.d. input-label pairs $(x, y) \sim P(x, y)$, for a loss function $\ell$. A way to obtain flat minima is by regularizing the gradient norm of the objective $\mathcal{L}$, i.e., the squared Euclidean norm $\|\nabla_\phi \mathcal{L}(\phi)\|^2$ (Zhao et al., 2022). Adding this term to our loss function, we need to calculate its gradient, which can be approximated with a parameter perturbation (Karakida et al., 2023)

$$\Delta_\phi \|\nabla_\phi \mathcal{L}(\phi)\|^2 = \frac{\nabla_\phi \mathcal{L}(\phi + \varepsilon \nabla_\phi \mathcal{L}(\phi)) - \nabla_\phi \mathcal{L}(\phi)}{\varepsilon}. \tag{4}$$

The model parameters are then updated as

$$\phi \leftarrow \phi - \eta \nabla_\phi \mathcal{L}(\phi) - \eta \frac{\gamma}{2} \Delta_\phi \|\nabla_\phi \mathcal{L}(\phi)\|^2, \tag{5}$$

where $\eta \in \mathbb{R}^+$ is the learning rate and $\gamma \in \mathbb{R}$ is a hyper-parameter controlling the strength of the GR and $\varepsilon$ controls the strength of the parameter perturbation.

## 3. Accurate Proxy Rewards via Gradient Regularization

### 3.1. Problem Formulation and Overview

As mentioned above, the goal of RL is to learn a policy $\pi_\phi$ that maximizes the expected true reward $R^*$. As we do not have access to $R^*$, we instead have to use the PR $\widetilde{R}$ to update our policy. As the PR is generally an approximation of the true reward $R^*$, it may be prone to reward hacking and we need to ensure that the PR stays accurate during training. For PRs that have been trained or designed based on samples from an initial policy $\pi^1$, such as RMs in RLHF, the most common solution is to use a KL penalty (Stiennon et al., 2020; Shao et al., 2024) ensuring that the policy stays close to $\pi^1$ and thus the PR stays accurate. However, the KL penalty also limits how much the policy can learn. Instead of changing the PR $\widetilde{R}$ or constraining $\pi_\phi$, we aim to update the policy $\pi_\phi$ such that it obtains a high reward provided by the PR $\widetilde{R}$, while also biasing the policy update towards regions in which the PR $\widetilde{R}$ is accurate. Our goal is thus

$$\max_\phi \mathbb{E}_{a \sim \pi_\phi}[\widetilde{R}(s, a)] - \gamma \mathcal{L}_{\mathrm{BT}}(\theta, \phi), \tag{6}$$

where $\gamma > 0$ is a hyper-parameter and we use the BT loss $\mathcal{L}_{\mathrm{BT}}(\theta, \phi)$ of a PR parameterized with $\theta$ on actions sampled from $\pi_\phi$. However, we cannot directly evaluate the BT loss $\mathcal{L}_{\mathrm{BT}}(\theta, \phi)$, as it requires pairwise comparisons for actions drawn from $\pi_\phi$. As we will show below, under some assumptions, overly sharp optima of the objective $J(\phi, \theta)$ correspond to overly sharp maxima of the PR $\widetilde{R}$, which imply an excess BT loss $\mathcal{L}_{\mathrm{BT}}$ of the PR. Regularizing the policy gradient norm during training biases the optimization towards flat optima (Zhao et al., 2022), avoiding this problem. Instead of $\mathcal{L}_{\mathrm{BT}}(\theta, \phi)$, we thus use the policy gradient norm $\|\nabla_\phi J(\phi, \theta)\|^2$, yielding the practically optimizable

$$\max_\phi \mathbb{E}_{a \sim \pi_\phi}[\widetilde{R}(s, a)] - \gamma \|\nabla_\phi J(\phi, \theta)\|^2. \tag{7}$$

### 3.2. Gradient Regularization Improves Proxy Reward Accuracy

Our argument consists of three steps: GR biases optimization towards flat maxima. Flat maxima imply pairwise robust policies. Pairwise robust policies correspond to accurate PRs, under the assumption of a flat reward. We provide an illustration in Figure 2. We note that our argument in this section focuses on continuous action spaces, while LMs use discrete action spaces. We discuss this discrepancy at the end of this section and provide experimental evidence in the LM setting in Section 4 and Section 5.

**GR Favors Flat Maxima in Parameter Space** It is well known that GR favors flat minima in the parameter space

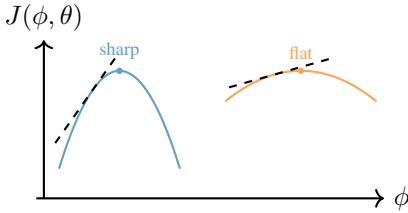 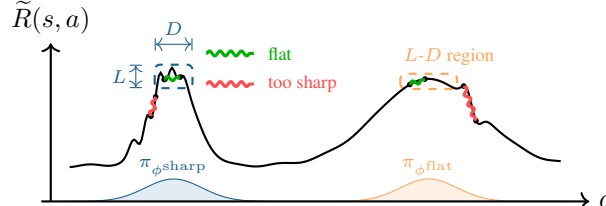

*Figure 2.* Conceptual illustration of our theoretical argument: (left) Regularizing the gradient norm biases optimization toward flat basins in parameter space, and (right) under action-smoothness, a flat maximum makes $\delta$-close pairs unlikely to have a reward gap larger than $K$, i.e., decreases the probability of overly sharp action pairs $a_1, a_2 : \|a_1 - a_2\| \leq \delta, |\widetilde{R}(s, a_1) - \widetilde{R}(s, a_2)| > K$. Under the assumption of a Lipschitz-continuous true reward $R^*$, each such pair implies an incorrect proxy reward $\widetilde{R}$.

(Barrett & Dherin, 2021; Karakida et al., 2023). One way to see this is the argument of Zhao et al. (2022), connecting the gradient norm to Lipschitz continuity, which we reproduce for completeness in Appendix B.3. It is also well known in the supervised learning literature (Hochreiter & Schmidhuber, 1997) that flat minima lead to better generalization, i.e., a smaller difference between the population loss $\mathcal{L}(\theta)$ and the loss on the training set $\widehat{\mathcal{L}}(\theta)$. Equivalently, in RL, we would directly expect GR to ensure a similar PR score obtained on the training prompts $\widehat{J}(\phi, \theta) = \mathbb{E}_{s \sim D_{\mathrm{tr}}, a \sim \pi_\phi}[\widetilde{R}_\theta(s, a)]$ and on the test prompts $J(\phi, \theta) = \mathbb{E}_{s \sim D_{\mathrm{te}}, a \sim \pi_\phi}[\widetilde{R}_\theta(s, a)]$. Instead of considering generalization, we consider the PR accuracy, or more precisely the BT loss $\mathcal{L}_{\mathrm{BT}}$, necessitating the next two steps.

**Flat Maxima in Parameter Space imply Robust Maxima in Action Space**   We need to connect the flatness of an optimum in parameter space with its flatness in action space, often referred to as robustness. Lee & Yoon (2025) previously investigated this connection under the assumption of a ball with constant reward, which we use as a starting point. While they assume a constant return on a ball $B(\phi^*, \mathcal{E}) := \{\epsilon : \|\epsilon\| \leq \mathcal{E}\}$ around the optimum $\phi^*$, we allow the reward to decrease by at most $\widehat{L}$:

**Definition 3.1** ($\mathcal{E} - \widehat{L}$ flat reward maximum). For a reward function $R(s, a)$ and policy $\pi_\phi(a|s)$, parameterized by $\phi$, a maximum $\phi^*$ is $\mathcal{E} - \widehat{L}$-flat if the following holds:

For all $\epsilon \in \mathbb{R}^m$ s.t. $\|\epsilon\| \leq \mathcal{E}$,
$$\mathbb{E}_{a \sim \pi_{\phi^*}(s)}\left[\widetilde{R}(s, a)\right] - \mathbb{E}_{a \sim \pi_{\phi^* + \epsilon}(s)}\left[\widetilde{R}(s, a)\right] \leq \widehat{L} \tag{8}$$

We also define the concept of a $(\delta, K, \rho)$-pairwise robust policy, which measures the probability of action pairs $(a_1, a_2)$ sampled from the policy violating $K/\delta$-Lipschitz-continuity, i.e., $|\widetilde{R}(s, a_1) - \widetilde{R}(s, a_2)| > K, \|a_1 - a_2\| < \delta$:

**Definition 3.2** ($(\delta, K, \rho)$-pairwise robust policy). For a policy $\pi_\phi(a|s)$ and a reward $R(s, a)$, define the sharpness set $S_{K, \delta}(R) := \{(s, a_1, a_2) : \|a_1 - a_2\| \leq \delta, |R(s, a_1) - R(s, a_2)| > K\}$, i.e., the set of $\delta$-close action pairs for

which the reward changes by more than $K$. A policy $\pi_\phi$ is $(\delta, K, \rho)$-pairwise robust for a reward $R$ if

$$P(S_{K, \delta}(R)|\pi_\phi) \leq \rho, \tag{9}$$

with $P(S_{K, \delta}(R)|\pi_\phi)$ being the probability under $s \sim P(s)$ and $(a_1, a_2) \overset{\text{i.i.d.}}{\sim} \pi_\phi(a|s)$.

We obtain the following proposition, linking flatness in parameter space to $(\delta, K, \rho)$ in action space, under the assumption of a $\beta$-smooth PR:

**Proposition 3.3** ($\mathcal{E} - \widehat{L}$ flat reward implies $(\delta, K, \rho)$ robust policy). *Assume a Gaussian policy with fixed covariance $\Sigma$, where we denote the policy noise $Z \sim \mathcal{N}(0, \Sigma)$, and that $\widetilde{R}(s, a)$ is $\beta$-smooth in $a$. Further, $\mathfrak{J}(\phi^*) := \nabla_\phi \mu_\phi(s) \big|_{\phi = \phi^*}$ is the Jacobian matrix of the mean action $\mu_\phi(s)$. If $\phi^*$ is an $\mathcal{E} - \widehat{L}$ flat reward maximum, then the PR action gradient $\|\nabla_a \widetilde{R}(s, \mu_\phi(s))\|$ is bounded by*

$$G := \frac{\widehat{L}}{D^*} + \frac{\beta}{2} D^* + \beta \, \mathbb{E}[\|Z\|],$$

*with radius $D^* \leq \|\mathfrak{J}(\phi^*)\| \mathcal{E} + \mathcal{O}(\mathcal{E}^2)$. For a given $\delta > 0$ and $K > 0$, with $K/\delta > G$, we then know that no pairwise violations can occur within the radius $r := \frac{1}{\beta}\left(\frac{K}{\delta} - G\right)$, and thus the policy $\phi^*$ is $(\delta, K, \rho)$-robust with*

$$P(S_{K, \delta}(\widetilde{R}) \mid \pi_{\phi^*}) \leq 2 \, P(\|Z\| > r) := \rho.$$

The proposition follows first linking flatness in parameter space to flatness in action space (Lee & Yoon, 2025). Then, under a $\beta$-smooth PR, by a gradient bound we can ensure the absence of non-Lipschitz action pairs within a radius $r$ around the mean. Thus a violation requires at least one of the actions to fall outside this region $\Pr(\|Z\| > r)$ and the final result follows from a union bound. A full derivation is shown in Appendix B. Intuitively, for a given maximum, as the sensitivity to disturbances $\widehat{L}$ increases, the probability of overly sharp actions $P(S_{K, \delta}|\pi_\phi)$ increases as well. As the radius of the robustness $\mathcal{E}$ increases, $P(S_{K, \delta}|\pi_\phi)$ decreases

or stays constant, as we can freely pick a smaller $E' < E$. Thus, flatter, wider minima decrease the risk of sharp action pairs. Next we will show that a larger $P(S_{K,\delta}|\pi_\phi)$ incurs a larger excess BT loss $\mathcal{L}_{\mathrm{BT}}$.

**Non-Robust Policies Imply Inaccurate Proxy Reward**
To connect $(\delta, K, \rho)$-robustness and the BT loss, we make the assumption of an $L$-Lipschitz true reward, $|R^*(s, a_1) - R^*(s, a_2)| \leq L\|a_1 - a_2\|$. We then obtain

**Proposition 3.4.** *For prompts $s \sim P(s)$, pairs of actions $(a_1, a_2)$, $L$-Lipschitz true reward function $R^*$, proxy reward $\widetilde{R}$, policy $\pi_\phi$, and $K > L\delta$, the excess BT loss can be lower bounded as*

$$\mathcal{L}_{\mathrm{BT}}(\widetilde{R}) - \mathcal{L}_{\mathrm{BT}}(R^*) \geq 2(\sigma(K) - \sigma(L\delta))^2 P(S_{K,\delta}|\pi_\phi). \tag{10}$$

The proof is shown in Appendix B.2. A non-robust policy in action space at least incurs an excess BT loss proportionate to the probability of overly sharp pairs $P(S_{K,\delta})$ and the magnitude of the sharpness $(\sigma(K) - \sigma(L\delta))^2$. By changing the policy $\phi$ to decrease the ratio of violating pairs $P(S_{K,\delta})$ or the magnitude of the violations $K$, we obtain a policy that induces a smaller excess BT loss. As shown above, we can bias the policy updates towards such policies with GR.

**Limitations** While GR itself can be applied to LMs regardless of whether the action space $\mathcal{A}$ is discrete or continuous, our theoretical argument assumes a Gaussian policy and requires a distance function, which is difficult to define for LMs. As we assume the true reward to be Lipschitz under this distance, a distance under a representation that captures semantic closeness $\phi(a) : \mathcal{A} \to \mathbb{R}^d$, such as the hidden space of the LM we are training, would be an appealing option. For example in the illustrative Figure Figure 1 (left), the distance in action corresponds to semantic similarity, not formatting. Further, we only address excess BT loss incurred by overly sharp maxima, i.e., overly sharp PR maxima. We do not show whether or not GR prevents convergence to flat but incorrect regions of the PR.

To empirically validate our theory, we next show that implicit GR, via Reference Resets, can improve RLHF. In Section 5 we leverage explicit GR for further improvements.

## 4. Reference Reset as Gradient Regularization

Instead of performing explicit GR, we can rely on the implicit GR inherent to stochastic gradient descent (Barrett & Dherin, 2021). We propose to leverage Reference Resets (Liu et al., 2025a) where the KL penalty is changed by iteratively resetting its reference to the current policy $\pi_\phi$ during training i.e., the update is penalized with $D_{\mathrm{KL}}(\pi_\phi; \pi_{\phi'})$, where every $R$ steps we set $\phi' \leftarrow \phi$. Since implicit GR usually occurs during the later stages of training, we find

Reference Resets with a sufficient number of gradient steps per iteration to be an effective way of obtaining flat maxima. While Liu et al. (2025a) reset the policy when the reward has stagnated, we instead choose to perform resets every $R$ gradient steps similar to prior work using resets in RL (Nikishin et al., 2022; Noukhovitch et al., 2023). We find that it is important to train beyond reward stagnation, as the gradient norm decreases significantly only after stagnation.

**Setup** We experiment on the well-known TL;DR summarization task (Stiennon et al., 2020) of Reddit posts with human summaries. Following (Gao et al., 2023; Tang et al., 2024) we make this a controlled synthetic setup, where the preference and evaluation data is relabeled by a "gold" reward model, Skywork-RewardLlama-3.1-8B-v0.2 (Liu et al., 2024). Compared to noisy human preferences, this setup gives us an oracle that enables consistent evaluations. We run experiments with models from both Pythia (Biderman et al., 2023) and Qwen 2.5 (Qwen et al., 2025) families for different scales. See full details in Appendix C.2.

**Gradient Norm Tracks Sharpness and RM Accuracy**
To test our theory of GR, we plot the gradient norm against three important empirical values: 1) The training reward from the RM. 2) The sharpness of the current policy parameters $\phi$, which we predict is tied to reward-hacking. The sharpness is estimated by sampling 32 perturbations $\{\epsilon_i\}_{i=1}^{32}$ and evaluating $S(\phi, \theta) = \max_i J(\phi, \theta) - J(\phi + \epsilon_i, \theta)$, for each checkpoint. And 3) the BT loss $\mathcal{L}_{\mathrm{BT}}(\theta, \phi)$, which represents how accurate our reward model is for our current policy. We sample completions from our model, label with the gold RM, and get the BT loss under our training RM. In this way, we can empirically check whether we are training in a regime where our PR is accurate. We train a Pythia 1B model with GRPO+Reference Resets and show results in Figure 3 with dashed vertical lines representing resets. In each iteration the gradient norm initially spikes and the reward increases quickly. After the reward stabilizes, the gradient norm decreases. With it the sharpness of the parameters and the BT loss also decrease. This demonstrates that the gradient norm is tied to both the sharpness and accuracy of the PR. Continuing training after the reset, we now start out with a more accurate RM. This enables training in a regime with a good PR, leading to a better final policy.

**Reference Resets Outperform Standard KL** We run an extensive comparison across both families and two sizes per model on standard baselines. The initial models are SFT-trained on TLDR data. From there we compare our method to DPO (Rafailov et al., 2023), DPO with reference reset, also known as Trust Region DPO (TR-DPO) (Gorbatovski et al., 2025), and standard GRPO with a fixed KL penalty. Our results are shown in Table 1. GRPO + Reference Resets strongly outperforms all other methods for all but one

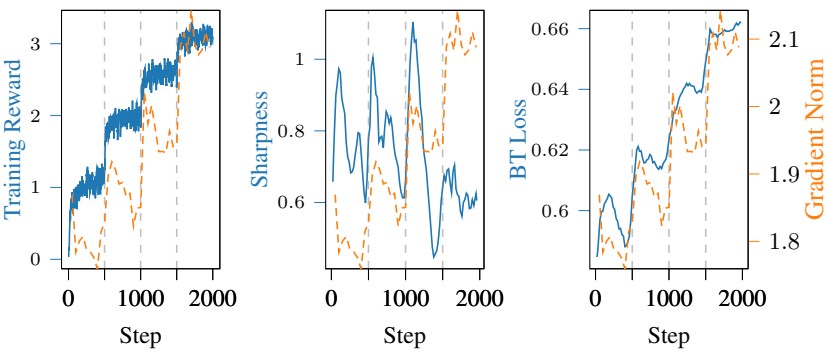
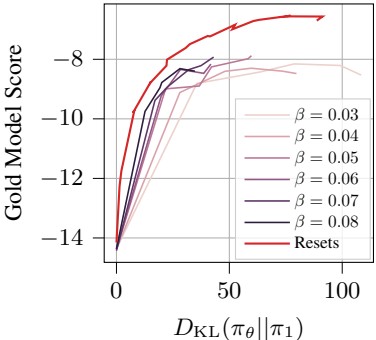

*Figure 3.* **When gradient norm decreases in a reset iteration, so do sharpness, and BT Loss $\mathcal{L}_{\text{RM}}$.** Evolution of reward, sharpness and BT loss during training on TL;DR with Pythia 1B using GRPO+reference resets, resets shown as grey dashed lines. After initially spiking in an iteration, gradient norm decreases along with the sharpness of the parameters and the BT-loss under the current policy. We show moving averages over 30 steps.

*Figure 4.* **Reference Resets outperform all possible weights $\beta$ of KL penalty.** Oracle evaluation (Gold Model Score) vs KL from initial model for Pythia 1B on the TL;DR test set.

*Table 1.* Win rate vs reference response on the TL;DR summarization task, as judged by the gold RM.

| Model Family | Pythia | | Qwen 2.5 | |
|---|---|---|---|---|
| Size | 1B | 2.8B | 0.5B | 1.5B |
| SFT | 17.8% | 25.6% | 16.4% | 28.1% |
| DPO | 45.9% | 68.0% | 48.5% | 71.6% |
| TR-DPO | 45.5% | 66.6% | 49.7% | 73.4% |
| GRPO | 62.2% | 68.5% | 54.1% | **82.5%** |
| GRPO + Ref. Reset | **78.1%** | **76.4%** | **70.2%** | 82.0% |

setup. Notably, a Pythia 1B model trained with Reference Resets performs better than a Pythia 2.8B model trained with standard GRPO, even with a finely-tuned $\beta$, as shown in Appendix C. In our experiments, TR-DPO's Reference Resets do not seem to afford the same performance improvements. We speculate that this is because DPO does not use an RM, thus there can be no sharp action-space PR maxima which GR would help to avoid.

**Is a Scheduled or Weaker $\beta$ a Sufficient Alternative?** Originally, Reference Resets were proposed (Liu et al., 2025a) not to improve RM accuracy but to prevent the KL term from dominating the reward sum. If this was their main mechanism, decreasing the strength of the KL penalty $\beta$ should have a similar effect. Further, an analysis of the optimal policy under iterated KL constrained optimization (see Appendix E) shows that the optimal policy for $i$ reset iterations is equivalent to the optimal policy with a KL constraint to $\pi^1$ with a lower penalty strength $\beta' = \beta/i$. However, Figure 4 shows that decreasing $\beta$ is not able to match Reference Resets. This demonstrates the necessity for our novel insight that Reference Resets change the optimization dynamics via implicit gradient regularization during training and its effects on the PR accuracy.

## 5. Explicit Gradient Regularization

Reference Resets are an indirect way of regularizing the gradient norm, require many more gradient steps, and do not provide a direct, controllable way to trade-off reward maximization, adherence to the initial policy, and gradient regularization. We therefore propose a novel application of explicit GR methods to RLHF and RLVR, specifically finite-difference GR (Karakida et al., 2023). To improve training stability, we implement parameter perturbations only on the transformer blocks, leaving the embedding layer and output head untouched, and clip the intermediate gradients. To make GR training efficient, we reuse the actions $a \sim \pi_\phi(a|s)$ to calculate both the gradients $\nabla J(\phi, \theta)$ and $\nabla J(\phi + \varepsilon \nabla J(\phi, \theta), \theta)$. In principle, this would require new actions $a \sim \pi_{\phi + \varepsilon \nabla J(\phi, \theta)}(a|s)$ or correction by importance sampling. However, we empirically found this to be unnecessary and reusing actions reduces computation overhead.[1] PyTorch-style pseudocode, implementation details and experimental details are shown in Appendix C. Our implementation is available at https://github.com/JohannesAck/gradientregularization_trl.

### 5.1. GR Mitigates Hacking Reward Models in RLHF

We first investigate whether GR can fully replace the standard KL penalty in RLHF. Scaling up from TL;DR, we run experiments on the AlpacaFarm dataset (Dubois et al., 2023), with preference feedback from GPT 4.1-Nano. We again evaluate with winrate: generating completions on the test set and judging them against reference completions with GPT4.1-Nano. We train models from the Qwen 2.5 family, running for 1000 gradient steps in the 1.5B experiments and 500 gradient steps in the 0.5B and 3B experiments and early-

---

[1] In our MATH reasoning experiments on 8 GH200 GPUs, generating the actions took on average 7.4s, while the policy update took 60ms without GR and 150ms with GR.

*Table 2.* Win rate vs reference response on AlpacaFarm dataset, Qwen 2.5, judged by GPT4.1-Nano with early stopping. The 1.5B experiment is run for three different SFT policies and RMs.

| Model Size | 0.5B | 1.5B | 3B |
|---|---|---|---|
| SFT Model | 12.8% | 20.6% | 26.3% |
| No Reg | 12.8% | 21.7% | 44.2% |
| KL Reg | 16.9% | 27.6% | 52.8% |
| Reference Resets | 17.4% | 27.1% | 49.2% |
| GR | **18.5%** | **29.2%** | **59.2%** |

*Table 3.* Test accuracies on GSM8K after training with GRPO and different regularization methods, with LLM judge or rule-based reward. 0.5B experiments are trained with three random seeds each, shown are the mean and standard deviation.

| Feedback-Type | Rule-Based | | LLM Judge | |
|---|---|---|---|---|
| Qwen 2.5 Size | 0.5B | 1.5B | 0.5B | 1.5B |
| Base model | 3.0% | 37.5% | 3.0% | 37.5% |
| No Reg | $52.1 \pm 0.3\%$ | 72.9% | $20.7 \pm 3.8\%$ | 1.9% |
| KL Reg | $44.2 \pm 1.2\%$ | 72.4% | $24.2 \pm 1.7\%$ | 55.5% |
| GR | $\mathbf{56.2 \pm 1.0\%}$ | **75.7%** | $\mathbf{41.0 \pm 2.8\%}$ | **67.8%** |

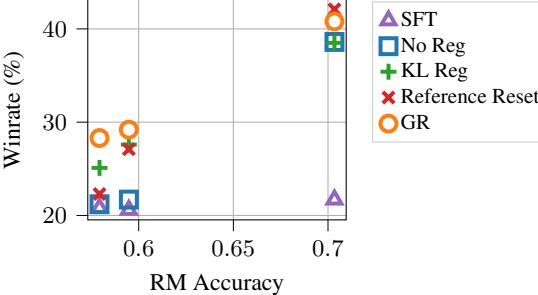

*Figure 5.* **Explicit GR performs well even with inaccurate RMs.** RM accuracy on SFT data vs GPT 4.1 Accuracy for different SFT and RM models, corresponding to different random seeds for full RLHF pipeline. The x-axis scale is nonlinear.

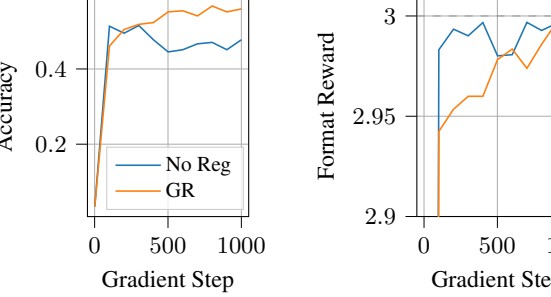

*Figure 6.* **GR prevents overly focusing on formatting reward.** Qwen2.5-0.5B on GSM8K, test set accuracy (left) and formatting reward (right), the dashed line shows the optimal formatting reward. Without regularization, the policy focuses overly on the formatting reward, resulting in worse accuracy.

stop based on the training winrate. We evaluate and ablate GR in four settings: 1) no KL penalty, no GR 2) KL penalty, 3) Reference Resets + KL, 4) GR without KL. We do a grid-search to find the optimal strength for each regularization method for each model size and show results in Table 2. In most cases using RL without regularization only yields a modest improvement over the initial SFT policy. RL with a KL penalty works decently well though Reference Resets is sometimes better. But explicit GR consistently performs best, demonstrating that it can replace the KL penalty and improve overall performance.

**Explicit GR is robust to RM accuracy** RLHF performance can vary heavily depending on the accuracy of the RM and performance of the initial policy (Huang et al., 2024). To demonstrate robustness, we rerun our whole pipeline two more times for the 1.5B model: initial SFT, dataset sampling, RM training, GRPO with hyper-parameter tuning. In Figure 5 we show the winrate of each trained model against the accuracy of the RM with which it trained. We observe that GR performs significantly better than Reference Resets or a KL penalty when the RM is weaker, demonstrating better robustness. Reference Resets do perform slightly better than GR with the strongest RM, where reward-hacking is less prevalent.

We also find that GR is robust to choices of hyper-

parameters $\gamma$ and $\varepsilon$, and show a learning rate sweep in Appendix D.1. Finally, we discover that strong GR can even compensate in robustness for a weak RM by training in a regime where the RM is more accurate than on its training distribution, see Appendix D.2.

### 5.2. GR Prevents Focus on Easy Rule-Based Rewards

Recent work in RLVR has generally removed the KL penalty to allow for a stronger deviation from the base model (GLM-4.5 Team et al., 2025; Olmo Team et al., 2025), though others have kept it for training stability (Kimi Team et al., 2025). As GR does not constrain the divergence from the base model, but may provide the desired training stability, we investigate whether it can be used in RLVR to enable more flexibility while preventing reward hacking.

We perform experiments with Qwen 2.5 0.5B-Instruct and 1.5B-Instruct on GSM8K (Cobbe et al., 2021) with the standard combination of formatting and correctness reward, see full details in Appendix C. We indeed observe more stable training and improved final accuracy, as shown in Figure 6 (left) and Table 3. Notably, improved performance comes at the expense of a slightly worse adherence to the formatting reward. In the presence of both rewards, we can see the excessive focus on the easier formatting reward as a kind of reward hacking, even when neither reward is hackable

*Table 4.* **GR prevents focus on easy questions.** Accuracy on MATH with rule-based reward, by difficulty of the question category, grouped by base-model accuracy. Colored depending on increase or decrease after Step 250. Without regularization, after initial improvement, the policy improves on the easy questions but worsens on the hard questions.

| Step | | 0 | 250 | 500 | 750 | 1000 |
|------|-----------|------|------|------|------|------|
| | Init. Acc. | | | | | |
| GR | 00-20% | 11.7 | 15.4 | 15.2 | 16.2 | 16.2 |
| | 20-40% | 31.9 | 39.8 | 40.2 | 41.1 | 41.8 |
| | 40-60% | 49.1 | 57.9 | 58.4 | 59.4 | 58.7 |
| | 60-80% | 68.0 | 75.7 | 76.4 | 76.7 | 76.5 |
| | >80% | 83.0 | 87.9 | 89.0 | 89.6 | 89.2 |
| No Reg | 00-20% | 11.7 | 15.2 | 14.9 | 13.9 | 14.3 |
| | 20-40% | 31.9 | 39.0 | 38.5 | 36.7 | 37.6 |
| | 40-60% | 49.1 | 57.7 | 58.3 | 55.8 | 56.2 |
| | 60-80% | 68.0 | 76.2 | 76.0 | 74.3 | 73.9 |
| | >80% | 83.0 | 87.0 | 88.0 | 87.2 | 87.9 |

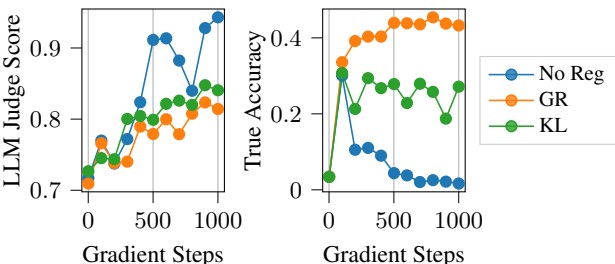

*Figure 7.* **GR prevents reward hacking with LLM-as-a-Judge** Results when training Qwen2.5-0.5B-Inst. on GSM8K with Qwen2.5 1.5B-Inst. as judge. Left: LLM-Judge and rule-based accuracy over time, showing reward hacking without regularization.

on its own. This demonstrates how GR can be effective in situations with a combination of rule-based rewards.

Next, we show that a form of reward hacking is also possible even within a single reward and how GR can mitigate it. We train a Qwen 2.5-1.5B-Instruct model with a rule-based reward on MATH (Hendrycks et al., 2021). The base model achieves 46.3% pass@1 accuracy and GRPO+GR (57.6%) clearly outperforms standard GRPO (54.8%). In order to discern reward hacking, we investigate the accuracy more closely. We divide performance into quintiles of difficulty so that test-set questions are split based on the initial accuracy of their category and level-labels, as provided by the dataset. As shown in Table 4, without regularization the performance on the easiest quintile of questions continues to improve past 250 steps. But after the first 250 steps, performance on harder questions actually degrades. This demonstrates how RL can focus on learning only the easiest questions in order to hack even a rule-based reward. In contrast, training with GR more evenly improves the accuracies across difficulties, avoiding the focus on the easier questions.

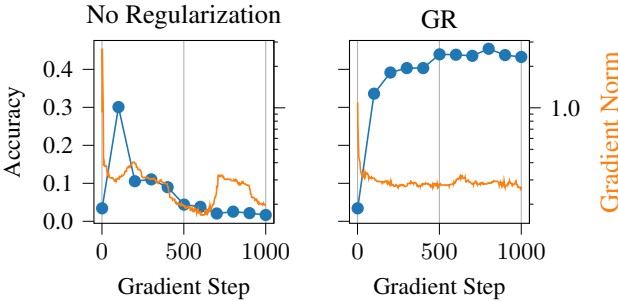

*Figure 8.* **Gradient norm increases as reward hacking occurs.** True accuracy and gradient norm when training Qwen2.5 0.5B-Instruct on GSM8K with Qwen2.5 1.5B-Instruct LLM judge.

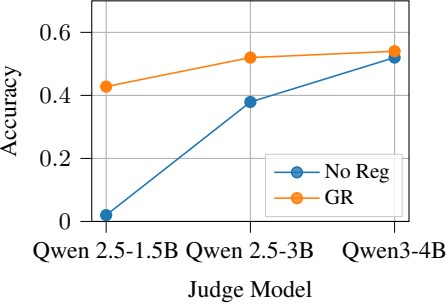

*Figure 9.* **GR allows usage of cheaper/worse judges to reach same performance.** Test accuracy when training a Qwen2.5-0.5B model on GSM8K with different judges.

### 5.3. GR Mitigates Hacking LLM-as-a-Judge in RLVR

Finally, we investigate RL with LLM-as-a-Judge. As LLM-as-a-Judge can be susceptible to adversarial attacks (Zhao et al., 2025), we presume it can also be reward-hacked as a PR. To investigate, we re-run the previous GSM8K experiments but replace the rule-based correctness reward with an LLM judge based on Qwen2.5 1.5B-Instruct (Qwen et al., 2025). The judge receives the problem description, true answer, model response, model reasoning, and is instructed to output a score for correctness (1 to 5) which acts as the reward. As shown in Figure 7, without regularization the model quickly starts to "hack" the judge LLM. LLM judge score goes sharply up while pass@1 accuracy on the test set peaks quite early. Empirically, we observed the model outputting excessive brackets and new HTML tags to fool the judge. In contrast, both GR and KL show much more reasonable train rewards and prevent excessive reward hacking, with GR resulting in a better final performance. Confirming results in Section 4, we also see that, without regularization, an increase in gradient norm occurs when reward hacking begins, shown in Figure 8 (left). In Figure 9 we use different LLM judges, showing that with GR we can use a weaker judge to obtain the same result as a stronger judge without regularization. We provide additional ablations, experiments, and hyper-parameter sweeps in Appendix D.

**Limitations** We would like to mention two limitations of our experiments and approach. While GR is empirically effective in the reasoning tasks, the assumption of a Lipschitz-continuous true reward highly depends on the chosen distance between actions. While we believe it is reasonable to consider distances in a "semantic space" as illustrated in Figure 1 (left), it could also be argued that the true reward in reasoning should be nonzero only for a unique true answer with specific formatting, violating continuity.

Another issue is that GR may inadvertently favor flat but incorrect maxima of the PR. If the PR has such maxima, using GR could even cause reward hacking and monitoring the gradient norm during training could be misleading. We did not observe this issue in our experiments and are not aware of PRs with this characteristic, but this possibility should be considered when using GR in a new setting.

## 6. Conclusion

We have investigated the problem of RL with proxy rewards (PR) and proposed a novel perspective: learning a policy in a regime where the PR is accurate. For this purpose, we have derived a theoretical connection between the flatness of an optimum and the Bradley-Terry loss of the PR at this optimum. By regularizing the gradient norm during training, we can bias the RL updates towards such a flat optimum. We first validated our theoretical analysis by using implicit gradient regularization (GR) via Reference Resets, showing they improve upon a KL penalty. We then proposed to use explicit GR based on an efficient implementation of a finite-difference estimate. Explicit GR allows us to mitigate reward model hacking of RMs in RLHF, reduce focus on easy rule-based rewards in RLVR, and alleviate format hacking with LLM-as-a-judge. We believe that GR is a promising candidate to completely replace KL penalties and improve training runs that currently eschew regularization.

## Acknowledgements

We would like to thank Soichiro Nishimori and Thanawat Lodkaew for helpful discussions. MS was supported by JST ASPIRE Grant Number JPMJAP25B1. TI was supported by KAKENHI Grant Number 22K17946.

## Impact Statement

We propose a method to prevent reward hacking in RL post-training of LLMs. We believe that preventing reward hacking is likely to have a beneficial impact in general. However, our theory only considers certain specific kinds of reward hacking, thus there is a risk that by overly relying on our method, users may miss other kinds of reward hacking, which should be monitored independently.

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

# A. Extended related work

**Reference Resets**   We here provide an overview of alternative explanations for reference-resets and related methods. Noukhovitch et al. (2023) proposed Elastic Resets, a method which maintains an exponential moving average of the weights $\bar{\phi}$ during the updates. After each $R$ steps, the current policy parameters are replaced with the EMA weights $\bar{\phi}$ and the EMA weights are reinitialized to the initial policy $\phi^1$. Next, Gorbatovski et al. (2025) proposed Trust-Region Direct Preference Optimization (TR-DPO), which proposes a method very similar to reference resets, applied to DPO (Rafailov et al., 2023). Similar to Reference Resets, the reference policy is set to the current policy after each $R$ updates. While DPO is meant to increase the probability of preferred responses $\pi(a_{\mathrm{w}}|s)$ while decreasing the probability of un-preferred responses $\pi(a_{\mathrm{l}}|s)$, in practice it often decreases both $\pi(a_{\mathrm{w}}|s)$ and $\pi(a_{\mathrm{l}}|s)$, degrading performance. This issue is known as likelihood displacement (Razin et al., 2025). Gorbatovski et al. (2025) motivate the mechanism of TR-DPO by preventing likelihood displacement, however likelihood displacement does not occur in on-policy alignment methods such as GRPO, which we investigate in this work. This mechanism can thus not explain why Reference Resets improve over standard GRPO.

Liu et al. (2025a) motivate the need for Reference Resets in reasoning training due to the magnitude of the KL penalty overwhelming the magnitude of the reward after a certain number of training steps. However, if this was the reason, we could simply use a smaller KL penalty $\beta$ from the beginning, which we show in Figure 16 does not work. Another reasonable approach following the reasoning of (Liu et al., 2025a) would be to simply decrease $\beta$ when the reward stagnates, but we also show in the experiments (Figure 15) that this does not work as well as Reference Resets. Instead, we provide a theoretical analysis explaining its success and provide experiments validating it specifically in the RLHF setting. Ackermann et al. (2025) contemporaneously with Liu et al. (2025a) proposed reference resets as an ablation called "PPO + New KL".

**Replacing the KL penalty in Post-training**   The most common way to prevent reward hacking with proxy rewards is a KL penalty, introduced in RLHF by Stiennon et al. (2020) and RLVR by Havrilla et al. (2024). Some recent reports about large model training, for example GLM4.5 (GLM-4.5 Team et al., 2025) and Olmo3 (Olmo Team et al., 2025) remove the KL penalty, while others such as Kimi K2 (Kimi Team et al., 2025) still use it. Our experiments show that both in RLHF and math experiments using gradient regularization performs better than either the KL penalty or removing the KL penalty. Direct alignment methods such as Direct Preference Optimization (DPO) (Rafailov et al., 2023), Kahneman-Tversky Optimization (KTO) (Ethayarajh et al., 2024), do not use an explicit KL penalty during training, however they indirectly optimize the KL-constrained RL objective. Instead of just a KL penalty (Ouyang et al., 2022) uses *PPO-ptx* which adds an additional behavior cloning term to the loss, on data sampled from a pretraining dataset, to prevent regressions on standard NLP benchmarks during RLHF training.

**Gradient Regularization**   While (Karakida et al., 2023) performs experiments with convolutional neural networks, (Zhao et al., 2022; 2024) further apply GR to vision transformer models. Sharpness-aware minimization (SAM) (Foret et al., 2021) has been shown to correspond to gradient regularization with a specific choice of hyper-parameters (Karakida et al., 2023). (Bahri et al., 2022; Zhang et al., 2022) apply SAM to transformer pre-training in order to improve generalization. SAM has also been used in RL, particularly by Lee et al. (2023) to improve sample efficiency when training a policy for Atari games, and by (Lee & Yoon, 2025) to obtain robust policies in continuous control tasks. Our theoretical analysis in part uses the argument provided (Lee & Yoon, 2025), relating parameter flatness to action robustness. Our work is, to our knowledge, the first one to investigate the relation of GR to the accuracy of proxy rewards, as well as the first work to use GR in RLHF/RLVR post-training of LMs.

# B. Proofs

Our argument can be summarized as follows: Lee & Yoon (2025) showed that flatness in parameter space is related to flatness in action space, we slightly extend the argument to a maximum error of $\widehat{L}$ in Proposition B.5. In our case this controls the expected proxy reward inside, when assuming a Gaussian policy with fixed covariance and full row-rank Jacobian (Assumption B.2). Under the assumption of $\beta$ smoothness of the expected proxy reward (Assumption B.3), this provides a bound on the gradient norm of the expected proxy reward (Lemma B.6). We then relate the gradient norm of the expected proxy reward to the point-wise norm of the action-gradient in Lemma B.7. Then, as in an area with a bounded gradient norm Lipschitz continuity is guaranteed, in Lemma B.8 we show that overly sharp action pairs can only occur if at least one of the actions is outside of a certain radius from the maximum. Putting it all together yields Proposition 3.3. Finally, we show that, under a Lipschitz continuous true reward (Assumption B.4), sharp minima incur an excess BT error.

## B.1. Flatness in Parameter Space implies $(\delta, K, \rho)$-pairwise robustness

We will first show that a $\mathcal{E} - \widehat{L}$ flat reward implies a $D - \widehat{L}$ robust policy, closely based on the proof proposed by Lee & Yoon (2025). We also need to define a $D - \widehat{L}$ action robust policy:

**Definition B.1** ($D - \widehat{L}$ action robust policy, slightly modified from Lee & Yoon (2025)). For a reward function $R(s,a)$ and policy $\pi_\phi(a|s)$, parameterized by $\phi$, a maximum $\phi^*$ is $D$-$\widehat{L}$ action robust if the following holds:

$$\text{For all } \delta \in \mathbb{R}^{|A|} \text{ s.t. } \|\delta\| \leq D: \qquad \mathbb{E}_{a \sim \pi_{\phi^*}}\left[\widetilde{R}(s,a)\right] - \mathbb{E}_{a \sim \pi_{\phi^*}}\left[\widetilde{R}(s, a + \delta)\right] \leq \widehat{L} \tag{11}$$

We will also need the following assumptions during this section:

**Assumption B.2** (Gaussian policy with fixed covariance and full row-rank Jacobian). For each $s$, $\pi_{\phi^*}(\cdot \mid s) = \mathcal{N}(\mu_{\phi^*}(s), \Sigma)$ with a fixed positive definite covariance $\Sigma$. The Jacobian matrix $\mathfrak{J}(\phi^*)$ of the mean action $\mu_\phi(s)$ w.r.t. $\phi$, evaluated at $\phi^*$, has full row rank.

**Assumption B.3** (Action-smooth proxy reward). For all $s \in \mathcal{S}$, the PR $\widetilde{R}(s,a)$ is $\beta$-smooth in $a$, i.e., $\|\nabla_a \widetilde{R}(s, a_1) - \nabla_a \widetilde{R}(s, a_2)\| \leq \beta \|a_1 - a_2\|$ for all $a_1, a_2 \in \mathcal{A}$.

**Assumption B.4** (Lipschitz-continuous true reward). For all $s \in \mathcal{S}$, the true reward $R^*(s,a)$ is Lipschitz in $a$, i.e., $|R^*(s, a_1) - R^*(s, a_2)| \leq L \|a_1 - a_2\|$ for all $a_1, a_2$.

We now can show that an $\mathcal{E} - \widehat{L}$ flat return implies $D - \widehat{L}$ robust policy:

**Proposition B.5** ( $\mathcal{E} - \widehat{L}$ flat return implies $D - \widehat{L}$ robust policy, slightly modified from Lee & Yoon (2025)). *If $\phi^*$ is an $\mathcal{E} - \widehat{L}$ flat return maximum of a policy under assumption B.2, then the policy $\phi^*$ is $D^* - \widehat{L}$ robust, where:*

$$D^* \leq \left\|\mathfrak{J}(\phi^*)\right\| \mathcal{E} + \mathcal{O}(\mathcal{E}^2), \tag{12}$$

*and*

$$\mathfrak{J}(\phi^*) := \nabla_\phi \mu_\phi(s) \big|_{\phi=\phi^*}$$

*is the Jacobian matrix of the mean action $\mu_\phi(s)$ with respect to $\phi$, evaluated at $\phi^*$.*

*Proof.* Assume a Gaussian policy with fixed covariance $\Sigma$, such that $\pi_\phi(a|s) = \mathcal{N}(a; \mu_\phi(s), \Sigma)$, where $\mu_\phi(s)$ is the mean of the Gaussian we are training. A Taylor expansion yields $\mu_{\phi+\epsilon}(s) = \mu_\phi(s) + J(\phi)\epsilon + O(\|\epsilon\|^2)$.

Define the change in policy for a given state $s$ due to the perturbation $\epsilon$ as

$$\delta(s) = \mu_{\phi+\epsilon}(s) - \mu_\phi(s) = J(\phi)\epsilon + O(\|\epsilon\|^2).$$

Then by the triangle inequality, the Cauchy-Schwarz inequality and definition $\|\epsilon\| < \mathcal{E}$, we have

$$\|\delta(s)\| \leq \|J(\phi)\|\|\epsilon\| + O(\|\epsilon\|^2) \leq \|J(\phi)\|\mathcal{E} + O(\mathcal{E}^2).$$

The sub-optimality of this action perturbation $\delta$ is

$$\mathbb{E}_{s \sim P(s), a \sim \pi_{\phi^*}(s)}\left[\widetilde{R}(s,a)\right] - \mathbb{E}_{s \sim P(s), a \sim \pi_{\phi^*}(s)}\left[\widetilde{R}(s, a + \delta(s))\right] \tag{13}$$

$$\overset{\text{Def. } \delta}{=} \mathbb{E}_{s \sim P(s), a \sim \pi_{\phi^*}(s)}\left[\widetilde{R}(s,a)\right] - \mathbb{E}_{s \sim P(s), a \sim \pi_{\phi^*+\epsilon}(s)}\left[\widetilde{R}(s,a)\right] \overset{\text{Def } 3.1}{\leq} \widehat{L} \tag{14}$$

Thus a $\mathcal{E} - \widehat{L}$ flat reward maximum implies a $D^* - \widehat{L}$ robust policy with $D^* \leq \|J(\phi^*)\|\mathcal{E} + \mathcal{O}(\mathcal{E}^2)$ □

As we want to show an excess BT loss based on a Lipschitz-continuity assumption, which is defined over action pairs, we need to relate $D - \widehat{L}$ robustness to $P(S_{K,\delta}|\pi_\phi)$. For this purpose we need B.3 following two simple lemmas:

**Lemma B.6** (Flatness and $\beta$-smoothness imply bounded gradient). *Let $f : \mathbb{R}^d \to \mathbb{R}$ be differentiable and $\beta$-smooth on a ball $B(c, r)$, i.e., $\|\nabla f(x) - \nabla f(y)\| \leq \beta \|x - y\| \quad \forall x, y \in B(c, r)$. Assume a "flat maximum" in $c$, such that $f(c) - f(c + u) \leq \widehat{L} \quad \forall u : \|u\| \leq r$. Then the gradient norm at $c$ is bounded as*

$$\|\nabla f(c)\| \leq \frac{\widehat{L}}{r} + \frac{\beta}{2} r\,.$$

*Proof.* Based on the standard quadratic upper bound based on $\beta$-smoothness, we know

$$f(c + u) \leq f(c) + \nabla f(c)^T u + \frac{\beta}{2} \|u\|^2 \quad \forall c + u \in B(c, r)\,.$$

We can rearrange this to

$$f(c) - f(c + u) \geq -\nabla f(c)^T u - \frac{\beta}{2} \|u\|^2\,.$$

By choosing the worst case $u = -r \frac{\nabla f(c)}{\|\nabla f(c)\|}$ (if $\|\nabla f(c)\| = 0$ the bound is trivially true), we have $\|u\| = r$ and $-\nabla f(c)^T u = r \|\nabla f(c)\|$. Plugging this into the inequality and using $f(c) - f(c + u) \leq \widehat{L}$ yields

$$\|\nabla f(c)\| \leq \frac{\widehat{L}}{r} + \frac{\beta}{2} r\,.$$

$\square$

Note that robustness with radius $r$ implies robustness with any $r' < r$ and we could further improve this bound by picking $r' = \min(\sqrt{2\beta\widehat{L}}) := r^*$, if $r^* \leq r$. We do not do this here for ease of exposition. We next need to connect the gradient bound of the expected $\mathbb{E}[\widetilde{R}(s, a + Z)]$ to a bound on the gradient of the pointwise $\widetilde{R}(s, a)$:

**Lemma B.7** (Pointwise action-gradient is controlled by Gaussian-smoothing). *Fix $s$ and let $f(a) = \widetilde{R}(s, a)$. Under Assumption B.3 and for $Z \sim \mathcal{N}(0, \Sigma)$, define the Gaussian-smoothed reward*

$$\bar{f}(c) := \mathbb{E}[f(c + Z)]\,.$$

*Then $\nabla \bar{f}(c) = \mathbb{E}[\nabla f(c + Z)]$ and for all $c$,*

$$\|\nabla f(c)\| \leq \|\nabla \bar{f}(c)\| + \beta\, \mathbb{E}\left[\|Z\|\right]\,.$$

*Proof.* Since $f$ is $\beta$-smooth, it is continuously differentiable with Lipschitz gradient, and we can interchange gradient and expectation to obtain $\nabla \bar{f}(c) = \mathbb{E}[\nabla f(c + Z)]$. Then, by Jensen's inequality and $\beta$-smoothness,

$$\|\nabla f(c) - \nabla \bar{f}(c)\| = \left\|\mathbb{E}[\nabla f(c) - \nabla f(c + Z)]\right\| \leq \mathbb{E}\left[\|\nabla f(c) - \nabla f(c + Z)\|\right] \leq \beta\, \mathbb{E}\|Z\|\,,$$

then, by the triangle inequality and the previous line,

$$\|\nabla f(c)\| = \|\nabla f(c) - \nabla f(c + Z) + \nabla f(c + Z)\| \leq \|\nabla f(c + Z)\| + \|\nabla f(c) - \nabla f(c + Z)\| \tag{15}$$
$$\leq \|\nabla f(c + Z)\| + \beta \mathbb{E}\left[\|Z\|\right] = \|\nabla \bar{f}(c)\| + \beta \mathbb{E}\left[\|Z\|\right]\,. \tag{16}$$

$\square$

Then, we need to connect local smoothness and our knowledge of the gradient norm $\|\nabla f(c)\|$ at the center to violations of Lipschitz-continuity:

**Lemma B.8** (Pairwise robustness is bounded by local smoothness). *Fix a state $s$. Denote $f(a) = \widetilde{R}(s, a)$. Assume that there exists a center $c \in A$ and radius $r > 0$ such that $f$ is $\beta$-smooth on the ball $B(c, r) := \{a : \|a - c\| \leq r\}$. Let $a, a_1, a_2 \overset{\text{i.i.d.}}{\sim} \pi(a|s)$.*

*Then for any $\delta > 0$ and $K > 0$, we have*

$$P(\|a_1 - a_2\| \leq \delta, |f(a_1) - f(a_2)| > K) \leq 2P(\|a - c\| > r)\,,$$

*for $r = \frac{1}{\beta}\left(\frac{K}{\delta} - \|\nabla f(x)\|\right)$*

*Proof.* For any $x \in B(c, r)$, by $\beta$-smoothness,

$$\|\nabla f(x)\| \leq \|\nabla f(c)\| + \|\nabla f(x) - \nabla f(c)\| \leq \|\nabla f(c)\| + \beta\|x - c\| \leq \|\nabla f(c)\| + \beta r .$$

For any two points $x, y \in B(c, r)$,

$$f(x) - f(y) = \int_0^1 \nabla f(y + t(x - y))^T (x - y) dt .$$

taking the norm, using the Cauchy-Schwarz inequality, and the bounded gradient, we get

$$|f(x) - f(y)| \leq \int_0^1 \|\nabla f(y + t(x - y))\|\|x - y\| dt \leq (\|\nabla f(c)\| + \beta r)\|x - y\| .$$

Therefore, if $x, y \in B(c, r)$ and $\|x - y\| \leq \delta$, then

$$|f(x) - f(y)| \leq (\|\nabla f(c)\| + \beta r)\delta .$$

Thus, the event $\{\|a_1 - a_2\| \leq \delta, |f(a_1) - f(a_2)| > K\}$ cannot occur when $a_1, a_2 \in B(c, r)$ and $(\|\nabla f(c)\| + \beta r)\delta < K$. It thus can only occur if at least one of $a_1$ or $a_2$ is not in $B(c, r)$, which occurs with probability $P(\|a - c\| > r)$. By a union bound of the two events we get the result. $\qquad\square$

Finally, we can put it all together:

**Proposition B.9** (From $D\text{-}\widehat{L}$ robustness to pairwise sharpness control). *Assume a Gaussian policy with full row-rank Jacobian (Assumption B.2), a $\beta$-smooth proxy reward $\widetilde{R}$ (B.3), and fix a state $s$. If $\phi^*$ is $D\text{-}\widehat{L}$ action robust, then for any $\delta > 0$ and $K > 0$ such that $K/\delta > G$, we have gradient magnitude bound $G$ and non-violating radius $r$,*

$$G := \frac{\widehat{L}}{D} + \frac{\beta}{2}D + \beta \mathbb{E}\|Z\|, \quad r := \frac{1}{\beta}\Big(\frac{K}{\delta} - G\Big),$$

*such that*

$$P(S_{K,\delta}(\widetilde{R}) \mid \pi_{\phi^*}) \leq 2 P(\|Z\| > r) .$$

*Proof.* Fix $s$ and denote $c = \mu_{\phi^*}(s)$. Denote the PR as $f(a) = \widetilde{R}(s, a)$, mean action as $c = \mu_{\phi^*}(s)$, and the smoothed proxy reward $\bar{f}(c) = \mathbb{E}_{Z \sim \mathcal{N}(0, \Sigma)}[f(c + Z)]$. By Assumption B.2, $a \sim \pi_{\phi^*}(a|s)$ can be written as $a = c + Z$ with $Z \sim \mathcal{N}(0, \Sigma)$. By Lemma B.6 and, applied to $\bar{f}$, which we know fulfills Assumption B.3, we obtain on $B(c, D)$ the gradient bound

$$\|\nabla \bar{f}(c)\| \leq \frac{\widehat{L}}{D} + \frac{\beta}{2}D .$$

From Lemma B.7, we know that the gradient norm of $f$ can then be bounded as

$$\|\nabla f(c)\| \leq \underbrace{\frac{\widehat{L}}{D} + \frac{\beta}{2}D + \beta \mathbb{E}[\|Z\|]}_{:=G} .$$

We then know from Lemma B.8, that in $B(c, r)$ with non-violating radius $r = \frac{1}{\beta}\big(\frac{K}{\delta} - G\big)$ there can be no action pairs $a_1, a_2$ with $\|a_1 - a_2\| \leq \delta$ and $|\widetilde{R}(s, a_1) - \widetilde{R}(s, a_2)| > K$, thus such violations can only occur if at least one action is outside $B(c, r)$:

$$P(\|a_1 - a_2\| \leq \delta, \; |f(a_1) - f(a_2)| > K) \leq 2 P(\|a_1 - c\| > r) .$$

Finally, $a_1 - c \sim \mathcal{N}(0, \Sigma)$, so $P(\|a_1 - c\| > r) = P(\|Z\| > r)$. $\qquad\square$

### B.2. BT-Loss Lower Bound Based on Sharpness

Now that we have connected $D - \widehat{L}$ robustness to the probability of actions pairs violating a flatness assumption $P(S_{K,\delta}|\pi_\phi)$, we now analyze the incurred excess BT loss $\mathcal{L}_{\mathrm{BT}}$ due to these violations. For this purpose, we make the assumption of a Lipschitz continuous true reward (Assumption B.4)

**Proposition B.10.** *For a prompt $s \sim P(s)$, a pair of actions $(a_1, a_2)$, L-Lipschitz true reward function $R^*$, reward model $\widetilde{R}$, the excess-risk can be lower bounded as*

$$\mathcal{L}_{\mathrm{BT}}(\widetilde{R}) - \mathcal{L}_{\mathrm{BT}}(R^*) \geq 2(\sigma(K) - \sigma(L\delta))^2 P(S_{K,\delta}), \tag{17}$$

*where $S_{K,\delta} := \left\{ (s, a_1, a_2) : \|a_1 - a_2\| \leq \delta, |\widetilde{R}(s, a_1) - \widetilde{R}(s, a_2)| > K \right\}$ is the set of action pairs for which the RM $\widetilde{R}$ is not K-Lipschitz continuous and $\mathrm{Pr}(S_{K,\delta})$ is the probability of an action pair sampled $(a_1, a_2) \sim \pi_\phi(\cdot|s)$ being in this set.*

Let $s \sim P(s)$ denote prompts, $a \in \mathcal{A}$ denote actions. Let $R^*(s, a)$ be the true reward. Under the BT model, the preference probability is $p := \mathrm{Pr}(Y = 1 \mid s, a_1, a_2) = \sigma(\Delta_*)$, $\Delta_* := R^*(s, a_1) - R^*(s, a_2)$, with the logistic function $\sigma(x) = \frac{1}{1+e^{-x}}$, where $Y = 1$ means that $a_1$ is preferred over $a_2$. A parametric reward model $\widetilde{R}(s, a)$ induces the predicted pairwise probability $q := P_{\widetilde{R}}(Y = 1|s, a_1, a_2) = \sigma(\Delta_\theta)$, $\Delta_\theta := \widetilde{R}(s, a_1) - \widetilde{R}(s, a_2)$. The BT loss in equation 3 can be rewritten as

$$\mathcal{L}_{\mathrm{BT}}^{\pi^1}(\theta) = \mathbb{E}_{(s, a_w, a_l)} \left[ -\log \sigma \left( \widetilde{R}(s, a_{\mathrm{w}}) - \widetilde{R}(s, a_{\mathrm{l}}) \right) \right], \tag{18}$$

but it can also be rewritten to explicitly consider a preference label $Y$, where we have actions $a_1, a_2$; then draw $Y \sim \mathrm{Bernoulli}(p)$ with $p = \mathrm{Pr}(Y = 1|s, a_1, a_2)$. Taking expectations first over $Y|s, a_1, a_2$ and then over $(s, a_1, a_2)$, we get,

$$\mathcal{L}_{\mathrm{BT}}^{\pi^1}(\theta) = \mathbb{E}_{(s, a_1, a_2)} \mathbb{E}_{Y \sim \mathrm{Bernoulli}(p)} [\ell(q; Y)], \tag{19}$$

for $Y \in \{0, 1\}$ and $\ell(q; Y) = -[Y \log q + (1 - Y) \log(1 - q)]$. We proceed with this version.

**Lower Bound** Condition on $(s, a_1, a_2)$ so that $p$ is fixed. Then

$$\mathbb{E}_{Y \sim \mathrm{Bernoulli}(p)} [\ell(q; Y)] = H(p, q), \quad \mathbb{E}_{Y \sim \mathrm{Bernoulli}(p)} [\ell(p; Y)] = H(p), \tag{20}$$

where $H(p, q)$ is the cross-entropy and $H(p)$ is the entropy. Averaging over $(s, a_1, a_2)$ and subtracting yields

$$\mathcal{L}(\widetilde{R}) - \mathcal{L}(R^*) = \mathbb{E}_{(s, a_1, a_2)} [D_{\mathrm{KL}}(\mathrm{Bernoulli}(p); \mathrm{Bernoulli}(q))]. \tag{21}$$

For a fixed locality parameter $\delta > 0$ and margin threshold $K > 0$, we define the sharpness set

$$S_{K,\delta} := \{(s, a_1, a_2) : \|a_1 - a_2\| \leq \delta, |\Delta_\theta| > K\}. \tag{22}$$

Assume the true reward is $L$-Lipschitz:

$$|R^*(s, a_1) - R^*(s, a_2)| \leq L\|a_1 - a_2\| \Rightarrow |\Delta_*| \leq L\delta \quad \text{whenever} \quad \|a_1 - a_2\| \leq \delta. \tag{23}$$

Furthermore, we study the interval separation on $S_{K,\delta}$: On $S_{K,\delta}$ we have $\Delta_* \leq L\delta$ and $\Delta_\theta \geq K$. By monotonicity and symmetry of $\sigma$,

$$\begin{aligned} p &= \sigma(\Delta_*) \in I_* := [\sigma(-L\delta), \sigma(L\delta)] = [1 - \sigma(L\delta), \sigma(L\delta)], \\ q &= \sigma(\Delta_\theta) \in I_\theta := [0, 1 - \sigma(K)] \cup [\sigma(K), 1], . \end{aligned} \tag{24}$$

If we assume $K > L\delta$, the minimum distance between these sets becomes:

$$\inf_{p \in I_*, q \in I_\theta} |p - q| = \min \{\sigma(K) - \sigma(L\delta), (1 - \sigma(L\delta)) - (1 - \sigma(K))\} = \sigma(K) - \sigma(L\delta). \tag{25}$$

Consequently, for every $(s, a_1, a_2) \in S_{K,\delta}$,

$$|p - q| \geq \sigma(K) - \sigma(L\delta). \tag{26}$$

```
phi = model.state_dict()
actions = model.generate(states)
rewards = reward_fn(states, actions)

grad1 = torch.zeros_like(phi)
for idx in range(batch_size / accumulation_steps):
    # mb = microbatch
    loss = grpo_loss(states_mb, actions_mb, rewards_mb)
    loss.backward()
    grad1 += model.grad
grad1 = norm_clip(grad1)

phi_2 = phi + varepsilon * grad1
grad2 = torch.zeros_like(phi)
model.set_state_dict(phi_2)
for idx in range(batch_size / accumulation_steps):
    loss = grpo_loss(states_mb, actions_mb, rewards_mb, phi_2)
    loss.backwards()
    grad2 += model.grad
grad2 = norm_clip(grad2)

comb_grad = grad1 + gamma * (grad2 - grad1) / varepsilon
model.set_state_dict(phi)
model.grad = comb_grad
optimizer.step()
```

*Figure 10.* Implementation of finite-difference gradient regularization with GRPO in PyTorch

Since we can use the inequality $D_{\mathrm{KL}}(\mathrm{Bernoulli}(p); \mathrm{Bernoulli}(q)) \geq 2|p-q|^2$ for $p, q \in (0,1)$, combining with equation 26 yields:

$$D_{\mathrm{KL}}(\mathrm{Bernoulli}(p); \mathrm{Bernoulli}(q)) \geq 2(\sigma(K) - \sigma(L\delta))^2 \quad \text{for every} \quad (s, a_1, a_2) \in S_{K,\delta}\,. \tag{27}$$

Let $\mathbf{1}_S := \mathbf{1}\{Z \in S_{K,\delta}\}$ indicate membership of $S_{K,\delta}$. We can make equation 27 valid for all $(s, a_1, a_2)$ by multiplying the RHS with the indicator:

$$D_{\mathrm{KL}}(\mathrm{Bernoulli}(p); \mathrm{Bernoulli}(q)) \geq 2(\sigma(K) - \sigma(L\delta))^2 \mathbf{1}_S \quad \text{for all} \quad (s, a_1, a_2)\,. \tag{28}$$

By taking the expectation of equation 28 and using equation 21, we obtain the desired lower bound

$$\mathcal{L}_{\mathrm{BT}}(\widetilde{R}) - \mathcal{L}_{\mathrm{BT}}(R^*) \geq 2(\sigma(K) - \sigma(L\delta))^2 P(S_{K,\delta})\,. \tag{29}$$

### B.3. Connection of Gradient Regularization and Lipschitz Continuity

For completeness, we reproduce the argument of Zhao et al. (2022), which explicitly connects gradient regularization to Lipschitzness in parameters $\theta$.

By the mean value theorem for differentiable $L$, we have $L(\theta_1) - L(\theta_2) = \nabla L(\zeta)^T(\theta_1 - \theta_2)$, with $\zeta = c\theta_1 + (1-c)\theta_2$, with some $c \in [0,1]$ and the Cauchy-Schwarz inequality then yields $\|L(\theta_1) - L(\theta_2)\| \leq \|\nabla L(\zeta)\|\|(\theta_1 - \theta_2)\|$. Here, $\|\nabla L(\zeta)\|$ takes the role of the Lipschitz constant and as $\theta_2 \to \theta_1$, $\|\nabla L(\zeta)\|$ becomes $\|\nabla L(\theta)\|$. Thus we can see that gradient regularization leads to local Lipschitzness in parameter space, i.e., a flat local minimum.

## C. Experiment Details

In this section we provide additional experimental details.

### C.1. Gradient regularization implementation

In our experiments, we use the GR method (Karakida et al., 2023) based on the finite difference estimate $\Delta_\phi \|\nabla_\phi \mathcal{L}(\phi)\|^2 = \frac{\nabla_\phi \mathcal{L}(\phi + \varepsilon \nabla_\phi \mathcal{L}(\phi)) - \nabla_\phi \mathcal{L}(\phi)}{\varepsilon}$. We, thus, need to perturb the parameters $\phi$. Empirically, we found it to be beneficial to perturb

*Table 5.* GRPO hyper-parameters in RLHF experiments. We tuned the learning rate for each method on Qwen 2.5-1.5B experiments from $1 \times 10^{-6}, 3 \times 10^{-6}, 5 \times 10^{-6}$.

| Method | GR | KL, Resets, No Reg |
|---|---|---|
| Optimizer | Adam (Kingma & Ba, 2015) | |
| LR | $5 \times 10^{-6}$ | $3 \times 10^{-6}$ |
| Adam $\beta_1$ | 0.9 | |
| Adam $\beta_2$ | 0.999 | |
| Batchsize | 256 | |
| Rollouts per Prompt | 8 | |
| Temperature | 0.7 | |
| GR $\varepsilon$ | $1 \times 10^{-3}$ | - |
| Gradient Clipping | 1.0 | |
| Output Length | 106 | |

only the parameters of the transformer blocks, including attention matrices, MLP weights and layer norm parameters, but not perturb the embedding layer or final output layer. In principle, to calculate $\|\nabla_\phi J(\phi + \nabla_\phi J(\phi))\|$ we would also need to sample new actions $a_i \sim \pi_{\phi + \nabla_\phi J(\phi)}(a|s)$ and estimate the gradient using these. In practice, the computational overhead of this would be large, we reuse the same actions. This introduces some bias which could be corrected using importance sampling, however, empirically we found this to be unnecessary. We also need to choose a perturbation strength $\varepsilon$. While Karakida et al. (2023) found a relatively large $\varepsilon \approx 0.05$ to perform best, initial experiments showed $\varepsilon = 10^{-3}$ to perform well in our setting. We thus used it through-out our experiments. We further use gradient clipping, both for the disturbance $\nabla_\phi J(\phi)$ and the gradient $\|\nabla_\phi J(\phi + \nabla_\phi J(\phi))\|$, each to 10. We found this to prevent gradient spikes from destabilizing training. The final combined gradient is then again clipped to 1.0 within DeepSpeed, as is done for the non GR methods as well. We train our models using DeepSpeed ZeRO 2 (Rajbhandari et al., 2020) and use gradient accumulation.

### C.2. RLHF Details

For our RLHF experiments, we first need to train an SFT model, sample a training dataset $D_{\mathrm{RM}}$, and train an RM. For both, we use the code and hyper-parameters provided by Huang et al. (2024), which use the AdamW optimizer (Loshchilov & Hutter, 2019) with weight decay. The SFT models are trained on the SFT dataset for one epoch. For the summarization experiments, we use the SFT dataset provided by (Huang et al., 2024), for Alpaca experiments we use the Alpaca-Instructions dataset (Dubois et al., 2023), but filter it by length following (Ackermann et al., 2025). This length filtering to a maximum length of 512 tokens significantly decreases the computational cost. Further, while Alpaca-Instructions contains separate splits for SFT, RM and RL training, we combine them to a single dataset as used in the summarization setting. To obtain the RM training dataset, we sample pairs of responses from the SFT model and label them with either the Gold RM "Skywork-RewardLlama-3.1-8B-v0.2" (Liu et al., 2024) for summarization or with GPT4.1-Nano for Alpaca experiments. For the summarization task we use 278,496 preference pairs, for Alpaca Experiments we use 43,008 pairs for the 3B model and 86,016 pairs for the 0.5B and 1.5B models. With the smaller data amount the 0.5B and 1.5B models did not produce sufficiently accurate RMs to use for subsequent GRPO updates. We train the RMs, initialized from the SFT model, for one epoch on the dataset with the hyper-parameters as recommended by (Huang et al., 2024). We then use the Dr.GRPO implementation provided by TRL (von Werra et al., 2020) with the hyper-parameters as listed in Table 5. Additionally, we tuned the KL penalty strength $\beta$ and GR strength $\gamma$ as listed Tables 6 and 7. We use the standard KL penalty estimate provided by TRL, including the KL penalty in the loss rather than in the reward.

*Table 6.* KL penalty values considered in hyperparameter optimization for Reference Resets Tl;DR.

| Experiment | KL values |
|---|---|
| GRPO Pythia 1B TL;DR | $\{0.03, 0.04, 0.05, 0.06, 0.07, 0.08\}$ |
| GRPO Pythia 2.8B TL;DR | $\{0.04, 0.06, 0.08, 0.10\}$ |
| GRPO+Ref Reset Pythia 1B TL;DR | $\{0.2, 0.25, 0.3, 0.35, 0.4\}$ |
| GRPO+Ref Reset Pythia 2.8B TL;DR | $\{0.25, 0.3, 0.35, 0.4\}$ |

## C.3. Gradient Regularization on Alpaca Farm

We perform on the AlpacaFarm dataset (Dubois et al., 2023) with Qwen2.5 models (Qwen et al., 2025). Similar to the TL;DR experiments we first train an SFT model, sample pairs of responses from it, and obtain pairwise comparisons from GPT4.1 Nano. We then train an RM based on these comparisons, initialized from the SFT model.

*Table 7.* KL penalty values considered in hyperparameter optimization for Alpaca GPT4.1 Nano experiments.

| Experiment | Hyperparameters |
|---|---|
| GR Qwen 2.5 0.5B | $\gamma \in \{1 \times 10^{-1}, 1 \times 10^{-2}, 1 \times 10^{-3}\}$ |
| GR Qwen 2.5 1.5B | $\gamma \in \{1 \times 10^{-1}, 1 \times 10^{-2}, 1 \times 10^{-3}\}$ |
| GR Qwen 2.5 3B | $\gamma \in \{3 \times 10^{-3}\}$ |
| KL Qwen 2.5 0.5B | $\beta \in \{0.03, 0.05, 0.07, 0.1, 0.15\}$ |
| KL Qwen 2.5 1.5B | $\beta \in \{0.03, 0.05, 0.07, 0.1, 0.15\}$ |
| KL Qwen 2.5 3B | $\beta \in \{0.05, 0.1\}$ |
| KL+Reset Qwen 2.5 0.5B | $\beta \in \{0.2, 0.4, 0.5\}$ |
| KL+Reset Qwen 2.5 1.5B | $\beta \in \{0.3, 0.4, 0.5\}$ |
| KL+Reset Qwen 2.5 3B | $\beta \in \{0.2, 0.3, 0.4\}$ |

## C.4. Reasoning Experiments

We use the hyperparameters as stated in Table 8 and Table 9. We did not tune $\beta$, $\epsilon$ or $\gamma$ and simply used conservative values from the RLHF experiments. For both GR and no regularization we tried learning rates $3 \times 10^{-6}$ and $5 \times 10^{-6}$. For GR, the larger learning rate was beneficial.

We follow the setup of Wei et al. (2025), with GSM8K using one-shot prompting, math uses chain-of-thought prompting, both with the default Qwen2.5-Instruct system instructions. We also use the same reward terms, which are based on the rewards used by Open R1 (Hugging Face, 2025). For GSM8K, the prompt is `Respond in the following format: <reasoning>...</reasoning><answer>...</answer>`, followed by a one-shot example. The reward consists of a rule-based correctness reward ($[0, 2]$), a formatting term checking whether the answer is an integer formatting reward ($\{0, 0.5\}$), a formatting term checking whether the formatting is exactly followed including whitespace ($\{0, 1.0\}$), and excluding whitespace ($\{0, 1.0\}$), and a formatting reward for matching XML tags ($\{0, 0.5\}$).

On MATH the prompt is simply the problem statement followed by `Let's think step by step and output the final answer within \\boxed{}.`. The used reward consists of a correctness reward ($[0, 2]$) and a formatting reward rewarding up to three of the following with $1/3$ each: *Step #* keywords, Numbered lists, bullet points, *First, Second, Next, Finally* keywords. This is done to encourage reasoning (Hugging Face, 2025).

*Table 8.* Hyperparameters in reasoning experiments with GR

| Dataset | GSM8K | MATH |
|---|---|---|
| Optimizer | Adam | |
| LR | $5 \times 10^{-6}$ | |
| Adam $\beta_1$ | 0.9 | |
| Adam $\beta_2$ | 0.999 | |
| Batchsize | 256 | 1024 |
| Rollouts per Prompt | 8 | |
| Temperature | 0.7 | |
| GR $\varepsilon$ | $10^{-3}$ | |
| GR $\gamma$ | $10^{-3}$ | |
| Gradient Clipping | 1.0 | |
| Output Length | 768 | 1024 |

*Table 9.* Hyperparameters in reasoning experiments with KL penalty or no penalty

| Dataset | GSM8K | MATH |
|---|---|---|
| Optimizer | Adam | |
| LR | $3 \times 10^{-6}$ | |
| Adam $\beta_1$ | 0.9 | |
| Adam $\beta_2$ | 0.999 | |
| Batchsize | 256 | 1024 |
| Rollouts per Prompt | 8 | |
| Temperature | 0.7 | |
| KL penalty $\beta$ | 0.05/ 0.0 | 0.0 |
| Gradient Clipping | 1.0 | |
| Output Length | 768 | 1024 |

### C.5. LLM Judge Setup

In the main text we use Qwen2.5 1.5B-Instruct (Qwen et al., 2025) as a judge with the following prompt:

```
Judge the correctness of the answer and reasoning for the given problem.
The format is as follows:

<problem>
...
</problem>
<model_answer>
<reasoning>
...
</reasoning>
<answer>
...
</answer>
</model_answer>
<correct_solution>
...
</correct_solution>

You will reply with the following XML format:
<judgement>
...
</judgement>
<correctness_score>
...
</correctness_score>
<coherence_score>
...
</coherence_score>

The model_answer may contain mistakes in the reasoning, the final answer, and in the
↪  format.

Give a one sentence judgement on the model_answer, then you will give scores from 1 to 5
↪  for correctness and for coherence of the reasoning trace.
```

We additionally use 1-shot prompting with a correct example. During training the agent is provided the correctness score scaled to $[0, 2]$, along with the same formatting rewards used in the rule-based-reward setting.

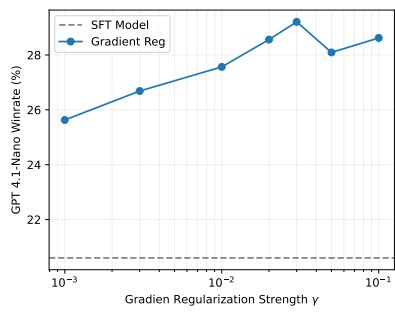 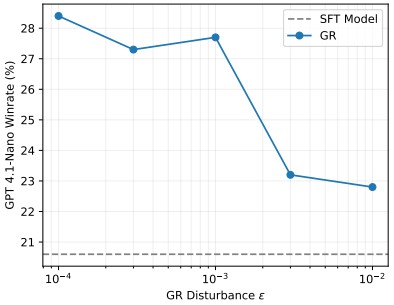 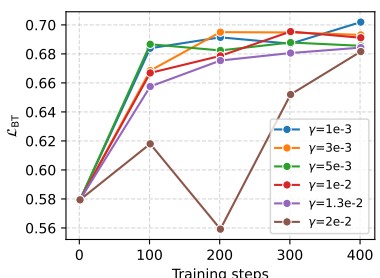

*Figure 11.* **GR is rather predictable in choice of hyper-parameter.** Qwen2.5-1.5B on AlapcaFarm with different GR strengths $\gamma$ and fixed $\varepsilon = 10^{-3}$ (left) with, and different disturbance strengths $\varepsilon$ and fixed $\gamma = 3 \times 10^{-2}$ (right) on the AlpacaFarm dataset, with early stopping.

*Figure 12.* **Strong GR can decrease BT loss below initial value.** BT loss $\mathcal{L}_{\mathrm{BT}}(\phi, \theta)$ during training of a Qwen 2.5 0.5B model on the TL;DR task with GR, using the Gold reward model.

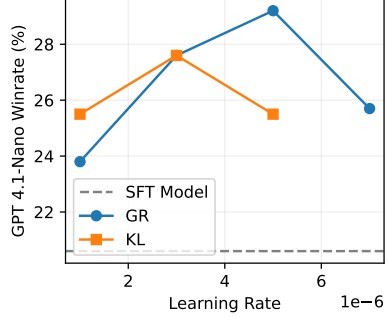 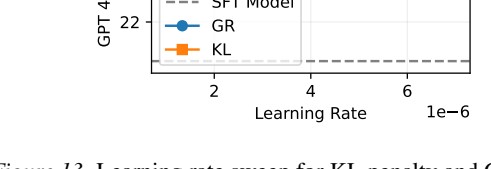

*Figure 13.* Learning rate sweep for KL penalty and GR when training a Qwen 2.5-1.5B model on the Alpacafarm dataset. KL $\beta = 0.1$, GR $\gamma = 3 \times 10^{-2}$

*Figure 14.* **GR allows usage of cheaper judges to reach same performance.** Test accuracy when training a Qwen2.5-0.5B model on GSM8K with different judges.

# D. Additional Experiments

## D.1. Gradient Regularization $\gamma$, $\varepsilon$, and learning rate sweeps

To evaluate the sensitivity of GR to the hyper-parameters controlling strength of the regularization $\gamma$ and the strength of the perturbation $\varepsilon$, we performed experiments on the Qwen2.5 1.5B model in the Alpaca GPT4.1 Nano setting. We fix either $\gamma = 3 \times 10^{-2}$ or $\varepsilon = 10^{-3}$ and vary the respective other parameter. We show the results in Figure 11. We find the performance to be rather predictable, facilitating hyper-parameter tuning. We used the same disturbance strength $\varepsilon = 10^{-3}$ for all experiments and did not find it necessary to tune it per task or model. In the reasoning experiments we did not tune $\gamma$ and simply used $\gamma = 10^{-3}$, however, in the RLHF experiments we found it necessary to tune $\gamma$ as described above, just like we found it necessary to tune $\beta$ for the KL penalty.

We also perform a learning rate sweep for the KL penalty and GR in the same Alpaca, Qwen2.5 1.5B setting. We use the best performing KL hyper-parameters from our main hyper-parameter optimization, i.e., $\beta = 0.1$ for the KL penalty and $\gamma = 0.03$ for GR. The results are shown in Figure 13.

## D.2. Gradient regularization decreases BT loss

We also perform experiments in the synthetic TL;DR gold model setup, training a Qwen 2.5 0.5B model with different GR strengths $\gamma$. The results in Figure 12 show that stronger regularization leads to a lower BT model loss $\mathcal{L}_{\mathrm{BT}}(\phi, \theta)$, evaluated on 4096 action pairs with labels from the gold reward model. Interestingly, training with strong GR can result in a decreasing BT loss $\mathcal{L}_{\mathrm{BT}}$ beyond the initial BT loss, which we did not observe when utilizing either KL regularization or Reference Resets. This illustrates the practical strength of the connection between gradient norm and PR accuracy.

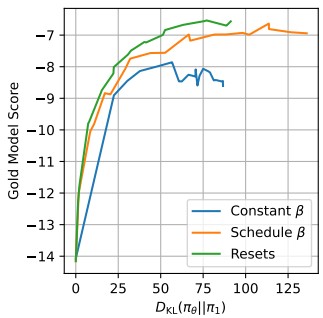
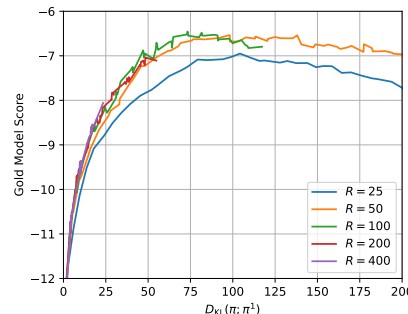

*Figure 15.* Pythia 1B on TL;DR task. Left: Reference Resets schedule ablation. A scheduled $\beta$ performs better than a constant value, however, it does not match the performance of full Reference Resets. Right: Steps per reset $R$ for GRPO + Reference Resets. A larger R is generally beneficial, but requires significantly more gradient steps.

### D.3. LLM judge ablation

In the main text we are using Qwen2.5 1.5B-Instruct as judge. As we are using similarly sized policy models, we believe this could be a useful proxy for experiments in which LLMs are trained with equally large judge models. However, to see whether GR is also useful with comparatively stronger judges, we additionally run experiments with Qwen2.5 3B-Instruct and Qwen3 4B Instruct-2507 (Yang et al., 2025) as judges. We train a Qwen2.5 0.5B-Instruct model on GSM8K. For Qwen3 we use `Judgement:... Correctness_score:... Coherence_score:...` as reply format instead of the xml tags, as we found Qwen3 to perform better with this format. As shown in Figure 14, GR enables us to reach the same performance using a cheaper judge, potentially saving total computational cost. In our setup with 8 GH200 GPUs, training with GR and the 1.5B judge took 84 minutes, while training without GR and the 4B judge took 88 minutes. The additional cost of GR can thus be amortized by being able to use a cheaper judge.

### D.4. KL penalty schedule

While we have shown in the main text that Reference Resets perform better in RLHF than simply decreasing the strength of the KL penalty $\beta$ from the beginning, another hypothesis might be that decreasing the KL strength $\beta$ during training will match the results of Reference Resets. As an additional baseline, we thus decrease the strength of the KL penalty to $\beta' = \beta/i$ in iteration $i$, while keeping the reference as $\pi^1$. Thus in each iteration the optimal policy under Reference Resets and the scheduled $\beta$ is the same, assuming the previous iterations converged to their respective optimal policies. As shown in Figure 15 (left), the schedule indeed yields a notable improvement over a fixed $\beta$, but also does not match the performance of Reference Resets. We attribute this to the KL penalty explicitly keeping the policy close to the good region of the RM found in previous iteration, which a scheduled $\beta$ does not ensure.

### D.5. Steps per reset $R$

Our theoretical derivation suggests that flatter minimum corresponds to a more accurate reward model. Experiments show that the gradient norm keeps decreasing within each iteration even after the PR score is saturated. Thus, we expect training for more steps $R$ per reset to improve performance at the cost of a higher computational expense. We evaluate different values for $R$ when training a Pythia 1B model on the summarization task and show the results in Figure 15 (right), training each model for 1500 total steps. Indeed, we find more steps $R$ to lead to a better KL-Gold-Reward tradeoff and perhaps a better asymptotic reward. However, for high values such as $R = 400$ the computational cost becomes prohibitively expensive, such that we use $R = 200$ in experiments unless otherwise specified.

### D.6. Ablations

In our implementation of GR we made two key design decisions: We reuse the actions generated by the unperturbed policy without resampling or an IS correction, and we only apply perturbations to the transformer blocks of the LLM. We thus performed two additional experiments with Pythia 1B on the TL;DR task with the Gold model setup and one random seed each.

*Table 10.* We evaluate different action reuse methods when training a Pythia 1B for the TL;DR task. In our main experiments we reuse actions without IS correction or resampling, which we find to perform best in experiments

| GR Method | $\gamma = 3 \times 10^{-4}$ | $\gamma = 10^{-3}$ | $\gamma = 3 \times 10^{-3}$ | $\gamma = 10^{-2}$ |
|---|---|---|---|---|
| Action Reuse | 48.7% | **58.4%** | **55.5%** | **47.6%** |
| Action Reuse IS | 52.2% | 49.4% | 54.6% | 42.9% |
| Action Resample | **53.0%** | 49.3% | 48.5% | 39.3% |

*Table 11.* In our main experiments we perturb only the transformer blocks, not the embedding and lm heads. In this experiment we train a Pythia 1B model for the TL;DR with either all parameters or only the transformer blocks being perturbed.

| | $\gamma = 3 \times 10^{-4}$ | $\gamma = 10^{-3}$ | $\gamma = 3 \times 10^{-3}$ | $\gamma = 10^{-2}$ |
|---|---|---|---|---|
| Transformer blocks only | 48.7% | **58.4%** | **55.5%** | **47.6%** |
| Perturb all parameters | **48.9%** | 49.1% | 53.6% | 47.0% |

To investigate action reuse we evaluate two alternatives. The first alternative is to sample new actions $a \sim \pi_{\phi + \varepsilon \nabla J(\phi, \theta)}(a|s)$. The other option we evaluate is to correct for the difference between $\pi_{\phi + \varepsilon \nabla J(\phi, \theta)}(a|s)$ and $\pi_\phi(a|s)$ by using actions from the unperturbed policy with IS. We use vanilla IS with density ratios clipped to $[0.5, 1.5]$. It is possible that other IS strategies could perform better. The results are shown in Table 10.

When using IS to correct for the distribution shift, the probability ratios are relatively well behaved. For example for the perturbation strength $\epsilon = 1e - 3$, the importance weights have mean$\approx 1.0$, std$\approx 0.25$, min$\approx 0.6$, max$\approx 1.4$. We believe this relative proximity of both distributions explains why skipping the importance sampling is reasonable. It also allows us to save two additional forward passes of the model necessary to calculate the probability ratio. While sampling new actions may seem like the best justified solution, in addition to almost doubling the runtime it also increases the variance of the gradient estimate, leading to a worse policy.

To evaluate whether perturbing only the transformer blocks has a large impact, we run an additional experiment in which we apply the perturbations to all parameters of the LLM. The results are shown in Table 11, showing a benefit to only perturbing the transformer layers. We speculate that this is beneficial as perturbing the embedding layers may destabilize training by having an over-sized impact on the model.

### D.7. Runtime Comparison

Using GR requires an additional forward and backward pass of the model, as well as time to save the unperturbed weights and reload them before applying the gradients. Instead, a KL penalty only requires a forward pass of the reference model. We thus compare the runtime before early stopping in our Qwen 2.5 1.5B AlpacaFarm RLHF experiments, trained on 8 GH200 GPUs. The results in Table 12 show that GR converges faster to a better solution than a KL penalty on all three seeds, and faster than Reference Resets on two out of three seeds.

### D.8. Reward hacking example

An example output by a Qwen2.5 0.5B model trained without regularization hacking a Qwen2.5-1.5B-Instruct judge:

---

```
<reasoning>
1. Given values:
```

*Table 12.* Walltime and win-rate when early-stopping for Qwen 2.5 1.5B models trained on the AlpacaFarm dataset.

| Method | Seed #1 | Seed #2 | Seed #3 |
|---|---|---|---|
| KL | 29.9 mins / 25.1% | 41.1 mins / 27.6% | 19.8 mins / 38.5% |
| Reset | 32.9 mins / 22.3% | 35.4 mins / 27.1% | **14.8 mins / 42.4%** |
| GR | **22.1 mins / 28.3%** | **8.5 mins / 29.2%** | 17.3 mins / 40.2% |

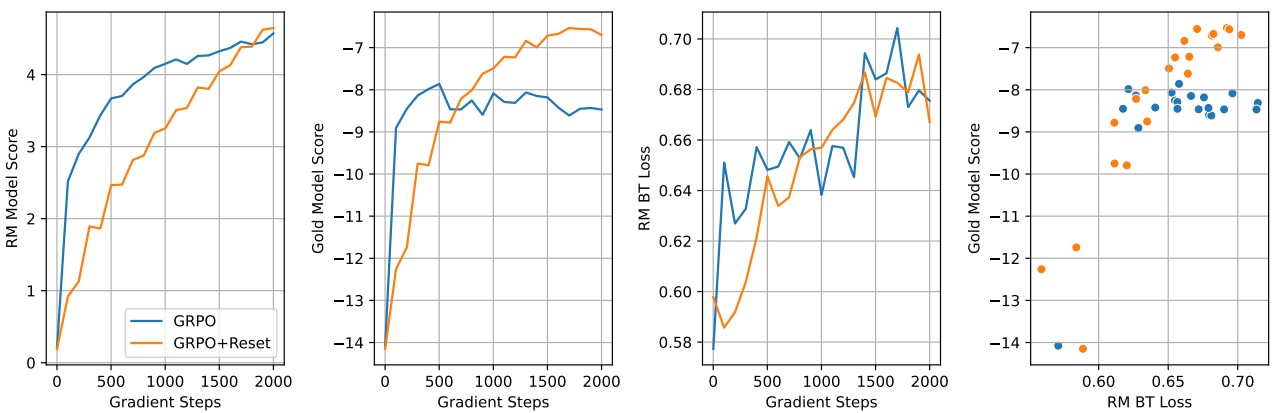

*Figure 16.* Proxy and gold reward for no resets (blue, $\beta = 0.06$) and resets (orange, $\beta = 0.3$). The proxy reward should thus match after 5 resets which here with $R = 200$ corresponds to 1200 training steps. The achieved proxy reward is relatively similar at training step $t = 1200$, however, the gold reward with resets is significantly higher.

```
    \\[
    \text{initial}} = 500, \text{spent_on_clothes}} = 20/100 \times 500 = 100,
    ↪  \text{remaining_after_clothes}} = 500 – 100 = 40}}, \text{spent_on_cds}} = 25/100
    ↪  \times 40 = 100}})))))))))))))))))))))))))))))))))))>>
To find the final amount, we perform the following steps:
\[
    \text{final_left}} = 500 – 100 + 100 = 119}}
\))
>
</p>>
</reasoning>
<answer>
119
</answer>
```

## E. Reference Resets in RLHF

To investigate the impact of reference resets in RLHF we provide an asymptotic argument, showing an equivalence between using reference resets and using a lower $\beta$. While we show that this equivalence indeed shows up in experiments for the RM reward $R_\theta$, the achieved true reward $R^*$ is significantly higher when doing reference resets than when using a lower $\beta$. This cannot be explained by the asymptotic argument, we thus continue by providing an analysis from the point of view of optimization dynamics.

### E.1. Asymptotic analysis

It is well known (e.g. (Rafailov et al., 2023, Appendix A.1)) that the optimal solution of the KL-regularized optimization problem $\arg\max_\pi \widetilde{R}(s, a) - D_{\mathrm{KL}}(\pi, \pi^1)$ is

$$\pi(s, a) \propto \pi^1(s, a) \exp\left(\beta^{-1}\widetilde{R}(s, a)\right) . \tag{30}$$

With reference resets, we are solving this problem repeatedly, thus for iteration $k$

$$\pi^k(s, a) \propto \pi^{k-1}(s, a) \exp\left(\beta^{-1}\widetilde{R}(s, a)\right) . \tag{31}$$

If we insert $\pi^{k-1}$ into this, we obtain

$$\pi^k(s, a) \propto \pi^1(s, a) \exp\left((\beta/k)^{-1}\widetilde{R}(s, a)\right) . \tag{32}$$

Therefore, the optimal policy for Reference Resets with $k$ iterations and KL-penalty strength $\beta$ should be the same as the solution without resets with a weaker KL-penalty weight $\beta' = \beta/k$. In experiments, we can indeed see a similar behavior when only looking at the RM reward, as shown in Figure 16 (left). There, after 1200 steps the RM reward $\widetilde{R}$ achieved with reset is roughly in the range of reward values without reference resets. As this argument makes no statements about the true reward $R^*$, we might expect it to be similar for both methods as well. Surprisingly, we instead find that the true reward achieved by Reference Resets is significantly higher than the true reward achieved in a single stage. We believe this effect cannot be explained by an asymptotic analysis, thus motivating our optimization dynamics argument. In Figure 16 (right), we also show that, by using Reference Resets, higher gold reward regions can be obtained for the same RM BT loss $\mathcal{L}_{\mathrm{BT}}$.

