# OpenReview forum: "Gradient Regularization Mitigates Reward Hacking in Reinforcement Learning from Human Feedback and Verifiable Rewards"
_ICML.cc/2026/Conference — ICML 2026 regular_

### Official Review · Reviewer_kCQr · 2026-03-09

**Soundness:** 4
**Presentation:** 4
**Significance:** 3
**Originality:** 3
**Overall Recommendation:** 5
**Confidence:** 4

**Summary:**

This paper studies reward hacking in reinforcement learning from human feedback (RLHF) and reinforcement learning with verifiable rewards (RLVR). It argues that reward hacking should be understood not primarily as excessive deviation from the initial policy, but as optimization toward sharp optima of a proxy reward, where the proxy becomes a less reliable stand-in for the true objective. Building on this view, the paper proposes biasing optimization toward flatter proxy-reward optima, motivated by a theoretical link between flatness and proxy reward accuracy under Bradley-Terry loss. The authors introduce gradient regularization (GR) on the policy gradient norm, interpret Reference Resets as an implicit form of GR, and develop a finite-difference GR that can be applied in both RLHF and RLVR. The empirical evaluation covers TL;DR summarization with a gold reward model, AlpacaFarm RLHF, GSM8K and MATH with rule-based rewards, and GSM8K with an LLM-as-a-Judge. Results across these settings suggest that GR consistently improves on standard KL regularization and is often competitive with, or better than, Reference Resets, while reducing multiple manifestations of reward hacking. Overall, the paper offers an interesting and timely optimization-based perspective on mitigating reward hacking beyond the standard KL-to-reference view.

**Compliance With Llm Reviewing Policy:**

Affirmed.

**Final Justification:**

I maintain my recommendation of Accept. The paper frames reward hacking as optimization toward sharp proxy-reward optima and introduces gradient regularization as a mitigation strategy, with thorough evaluation across RLHF and RLVR settings. The rebuttal resolved my concerns: the added ablations demonstrate robustness to implementation choices, and the authors clarified the theory-experiment relationship. The paper makes a solid contribution to understanding and mitigating reward hacking.

**Key Questions For Authors:**

1. The theory is built on continuous actions, Gaussian policies, smooth proxy rewards, and a distance notion over actions, while the experiments are on discrete autoregressive LMs. Can the authors better clarify which parts of the observed LM behavior they believe are directly predicted by the theory, and which parts are currently supported only empirically?

2. Explicit GR depends on several implementation choices, including reusing sampled actions under perturbed parameters, perturbing only transformer blocks, and gradient clipping at multiple stages. How sensitive are the results to these choices, and can the authors provide ablations that isolate their impact?

3. In the RLHF setting, evaluation is based mainly on a gold RM or GPT-4.1-Nano judgments rather than direct human evaluation. Do the authors have evidence, even on a smaller scale, that the improvements from GR transfer to human preference judgments?

**Limitations:**

The paper does discuss the gap between the continuous-action theory and discrete LMs, and it also notes that GR only addresses certain forms of reward hacking. However, I think the limitations section could more explicitly discuss the additional computational overhead of explicit GR and the possibility that GR may still favor flat but semantically misaligned proxy-reward regions.

**Strengths And Weaknesses:**

This paper is a solid contribution that advances our understanding of reward hacking and suggests a promising direction for mitigation. The perspective is interesting: rather than viewing reward hacking mainly as excessive divergence from the initial policy, the paper frames it as optimization toward sharp proxy-reward optima where the proxy becomes unreliable. Given the importance of reward hacking in RLHF and RLVR, this perspective feels broader than a single benchmark and constitutes a meaningful conceptual contribution. The paper supports this perspective with both theoretical analysis and broad empirical evaluation across RLHF, rule-based RLVR, and LLM-as-a-Judge settings. The empirical results are generally convincing and suggest that gradient regularization (GR) can mitigate several forms of reward hacking more effectively than standard KL regularization. The paper is also generally well presented. It is clearly structured, and the progression from the problem formulation to the theory, implicit GR interpretation, and explicit GR method is easy to follow. The novelty comes from connecting flatness to proxy reward accuracy, reinterpreting Reference Resets as implicit GR, and adapting explicit finite-difference GR to RLHF/RLVR with practical implementation details and broad experimental support.

The main limitation is the gap between the theory and the actual language model setting. The theoretical analysis is developed under continuous-action assumptions with Gaussian policies, while the experiments are on discrete autoregressive LMs. The paper is transparent about this, but the theory should still be interpreted more as a useful explanation than as a direct guarantee for LM optimization. A second limitation is that the explicit GR method depends on several practical design choices, such as partial parameter perturbation and action reuse approximations, whose individual contributions are not fully isolated. Overall, I find the paper sound and potentially impactful, though these issues keep me from being more enthusiastic.

---

> ### Author Rebuttal · Authors · 2026-03-30
>
> We would like to thank the reviewer for their helpful comments.
>
> > Q2 Ablations
>
> We ran ablations for the partial perturbations and action reuse on Pythia 1B in the Gold reward model TL;DR setting, with a reduced budget of 500 timesteps and early-stopping:
>
> For different action reuse methods, perturbing only the transformer blocks:
>
> | GR Method       | $\gamma$=3e-4 | $\gamma$=1e-3 | $\gamma$=3e-3 | $\gamma$=1e-2 |
> |-----------------|---------------|---------------|---------------|---------------|
> | Action Reuse    | 48.7%         | **58.4%**         | **55.5%**         | **47.6%**         |
> | Action Reuse IS | 52.2%         | 49.4%         | 54.6%         | 42.9%         |
> | Action Resample | **53.0%**         | 49.3%         | 48.5%         | 39.3%         |
>
> For different perturbation methods, using action reuse without IS:
>
> |                                                          | \gamma=3e-4 | \gamma=1e-3 | \gamma=3e-3 | \gamma=1e-2 |
> |----------------------------------------------------------|-------------|-------------|-------------|-------------|
> | Transformer blocks only                                  | 48.7%       | **58.4%**       | **55.5%**       | **47.6%**       |
> | Perturb all parameters  | **48.9%**       | 49.1%       | 53.6%       | 47.0%       |
>
> Overall, we find that action reuse and perturbing only the transformer blocks performs best.
> Here we used vanilla IS with ratios clipped to [0.5,1.5]. It is possible that a more careful choice of IS method, such as flattened importance weights or better clipping, could improve the results.
>
> > Q1 Which behavior is predicted by theory, which is not.
>
> The theory only predicts that we will avoid sharp minima, which we know to imply inaccurate proxy rewards. We believe this strongly applies in the case of the RLHF experiments with trained reward models, as well as in the case of the LLM-as-a-judge setting. The rule-based reward reasoning experiments are more empirical, especially the result of GRPO without GR focusing on easy questions in MATH while also improving on the hard questions when using GR. Another part we do not consider in the theory is the capability of GR to improve generalization independent of reward hacking. We do not believe this to play a large role in the LLM-as-a-judge and RLHF experiments, as we see clear reward hacking there, but it could be a contributor in the rule-based tasks.
> We will add this to the discussion of the experiments, thank you!
>
> > Q3 Transfer to human judgements
>
> As shown by [Zheng et al (2023)](https://arxiv.org/pdf/2306.05685), we believe that training with LLM-as-a-judge can generalize to human judgements, so we expect GR to transfer well. Learning directly from human preferences is difficult for research setups as annotators can disagree with each other, the [original TLDR paper](https://arxiv.org/abs/2009.01325) notes an inter-annotator agreement of about 67%, leading to a more noisy learning signal.
>
> > computational overhead of explicit GR
>
> While each training step is more expensive, in the response to the reviewer G1in we are showing that GR often achieves a better reward earlier than a KL penalty, when using early stopping. We will add this to the draft.
>
>
> > GR may still favor flat but semantically misaligned proxy-reward regions
>
> We mention this in the limitations paragraph and the impact statement, but we agree that it should be discussed more fully and will extend it.
>
> Thank you for the review, we believe that the paper has been improved a lot with the recommended changes!

---

> > ### Author Rebuttal · Reviewer_kCQr · 2026-04-02
> >
> > Thank you for the rebuttal. The added ablations address my concern about implementation sensitivity, and the clarification of the theory versus empirical evidence improves the paper's positioning. The discussion of compute overhead is also helpful. Overall, the rebuttal resolves my main concerns, and I encourage the authors to reflect these clarifications in the revision.

---

> > > ### Author Response · Authors · 2026-04-04
> > >
> > > We would like to again thank the reviewer for their helpful recommendations.
> > >
> > > We plan to include the additional discussion of the theory and experimental setting, as well as the new experiments in a revised version of our work.

---

### Official Review · Reviewer_iYnW · 2026-03-11

**Soundness:** 3
**Presentation:** 4
**Significance:** 3
**Originality:** 2
**Overall Recommendation:** 5
**Confidence:** 2

**Summary:**

This paper investigates robustness of LMs trained via RLHF or RLVR against reward hacking via the lens of gradient regularization (GR). After theoretically and empirically (through implicit GR) motivating the relevance of GR for RLFHF/RLVR, an explicit GR approach is proposed and evaluated on some exemplary RLHF/RLVR tasks. The results indicate that GR is a suitable approach to improve overall performance and there is emprical support pointing at the direction that the proposed approach indeed mitigates reward hacking.

**Compliance With Llm Reviewing Policy:**

Affirmed.

**Final Justification:**

My concerns have been addressed and I maintain score to accept.

**Key Questions For Authors:**

1. do the authors have any intuition or support for the reuse of actions works (first paragraph of Section 5)? E.g. when performing IS, is there anything of interest about the weights?

**Limitations:**

yes

**Strengths And Weaknesses:**

* **Soundness**: The paper presents a thorough argument through theory and empirical analysis using implicit GRs. Although experiments on additional RLHF/RLVR datasets and tasks could be performed to better support the claims, the selection is reasonable and valid. The formulation of some of the claims is a bit strong to me, especially in the title "prevents" is a bit overconfident considering the marginal performance gains. If this is allowed by the ACs, I would argue for changing the title to something a bit more modest. Additionally, there is no measure of spread (although the authors seemed to have ran for multiple seeds according to Fig. 3, which is laudable) and there are no statistical tests to assess whether the differences between e.g. KL Reg or implicit GR via reference resets is significant (although if this were not the case, the conceptual/theoretical contributions would still maintain their value).
* **Presentation:** The paper is generally exceptionally well organised and written, and figures are intuitive, clear and supportive of the main points of the paper. One grip I have is with the choice for a scatterplot in Figure 5 which I think is not informative. There are some minor inconsistencies and nitpicks in formatting but these should be easy to fix in a camera ready.
* **Significance:** The addressed problem is relevant and, although the empricial gains over the state-of-the-art are somewhat limited considering this is a more theoretically grounded explicit approach to what was already being done in practice implicitly, I view the conceptual contributions are warranting acceptance in ICML. The paper is also timely considering the popularity of LMs and RLHF/RLVR.
*  **Originality/Novelty:** The paper mostly draws from existing work on e.g. GR and reward robustness, the theoretical contributions stemming from the contribution are novel.

### detailed remarks
* please use different markers/line styles for plots to accomodate color blindness
* the TL;DR task spelling is inconsistent, it would be useful to harmonize this
* the bibliography contains many arxiv URLs, for works that are past the preprint stage. Please refer to the official proceedings as much as possible
* ln 153/154: "optimization toward to flat maxima" < typo
* ln 188/189: there seems to be a spurious whitespace after "Proposition 3.3 ("
* ln 254: "Figure Figure 1"
* ln 259: I think "demonstrates" is a bit strong for the provided evidence
* ln 382/383: "for a weak RM by training in **a** regime"

---

> ### Author Rebuttal · Authors · 2026-03-30
>
> We would like to thank the reviewer for their helpful comments.
>
> >Q1: Do the authors have any intuition or support for the reuse of actions works (first paragraph of Section 5)? E.g. when performing IS, is there anything of interest about the weights?
>
> When using IS to correct for the distribution shift, the ratios are relatively well behaved (for \eps=1e-3, mean ~1.0, std ~0.25, min ~0.6 , max ~1.4). We believe this relative proximity of both distributions explains why skipping the importance sampling is possible. It also allows us to save two additional forward passes of the model.
> While sampling new actions may seem like the best justified solution, in addition to almost doubling the runtime it also increases the variance of the gradient estimate, leading to a worse policy.
>
> For completeness, we ran an ablation comparing IS, sampling new actions and simply reusing the actions (as done throughout the submission) on Pythia1B for TL;DR, with early stopping and a reduced maximum of 500 steps.
> The results are shown in the table below:
>
> | GR Method       | $\gamma$=3e-4 | $\gamma$=1e-3 | $\gamma$=3e-3 | $\gamma$=1e-2 |
> |-----------------|---------------|---------------|---------------|---------------|
> | Action Reuse    | 48.7%         | **58.4%**         | **55.5%**         | **47.6%**         |
> | Action Reuse IS | 52.2%         | 49.4%         | 54.6%         | 42.9%         |
> | Action Resample | **53.0%**         | 49.3%         | 48.5%         | 39.3%         |
>
> We clipped the importance sampling ratios to [0.5,1.5], it is possible that a more careful choice of IS method could improve results.
> We will add the ablation and discussion to the paper, thank you!
>
> > multiple seeds
>
> We ran three additional seeds for the Qwen2.5-0.5B rule-based and LLM-Judge GSM8K experiments, showing a consistent increase in performance:
>
> Rule-based:
>
> | Method | Seed #1 | Seed #2 | Seed #3 | Avg + STD  |
> |--------|---------|---------|---------|------------|
> | No Reg | 51.9%   | 52.5%   | 52%     | 52.1%±0.3% |
> | KL     | 43.5%   | 45.6%   | 43.6%   | 44.2%±1.2% |
> | GR     | **56.7%**   | **55.0%**   | **56.8%**   | **56.2%±1.0%** |
>
> LLM-Judge:
>
> | Method | Seed #1 | Seed #2 | Seed #3 | Avg + STD  |
> |--------|---------|---------|---------|------------|
> | No Reg | 24.0%   | 16.5%   | 21.6%   | 20.7%±3.8% |
> | KL     | 23.4%   | 23.1%   | 26.1%   | 24.2%±1.7% |
> | GR     | **37.9%**   | **41.8%**   | **43.4%**   | **41.0%±2.8%** |
>
> As the reviewer noted, we already show different random seeds for the main RLHF setting in Figure 3, showing that GR reliably performs better than a KL penalty here. We believe the consistent improvements across different settings indicate that GR performs well reliably.
>
>
>
> > detailed remarks
>
> We would like to thank the reviewer for their remarks, we will update the links in the biography, change the markers/colors to be colorblind-friendly and of course also fix the typos. Thank you!

---

> > ### Author Rebuttal · Reviewer_iYnW · 2026-04-01
> >
> > Thanks for clarifying some of these issue. My first final remaining concern is with the measure of spread across results, and lacking statistical significance tests. Can you please provide some insights into the significance of the results?
> >
> > My second final concerns is with the strong claims of "preventing" reward hacking in the title, and kindly request the authors to reconsider the title formulation or better support this claim by showing that reward hacking under the proposed method no longer occurs.

---

> > > ### Author Response · Authors · 2026-04-03
> > >
> > > Thank you for the rebuttal acknowledgement.
> > >
> > > > measure of spread across results, and lacking statistical significance tests
> > >
> > > To quantify statistical reliability, we extended the Qwen 2.5 0.5B LLM-Judge experiments above to ten seeds for each method:
> > >
> > > | Method    | No Reg     | KL         | GR         |
> > > |-----------|------------|------------|------------|
> > > | Avg + STD | 16.5%±7.8% | 24.6%±1.7% | 40.5%±2.7% |
> > >
> > > A one-sided Welch's t-test, assuming normality, provides the following p values
> > >  * GR > KL: p<0.001
> > >  * GR > No Reg: p<0.001
> > >  * KL > No reg: p=0.042
> > >
> > > These results indicate that GR indeed provides a reliable improvement over both a KL penalty and not using any regularization.
> > >
> > > > preventing" reward hacking in the title
> > >
> > > If possible, we will change the title to "mitigates" rather than "prevents". Small changes to titles were possible under last year's camera ready guidance, so we believe this should be possible.
> > >
> > > Again, thank you for your review!

---

### Official Review · Reviewer_G1in · 2026-03-13

**Soundness:** 2
**Presentation:** 2
**Significance:** 3
**Originality:** 3
**Overall Recommendation:** 3
**Confidence:** 3

**Summary:**

This paper proposes gradient regularization (GR) as a general mechanism to prevent reward hacking in RL post‑training of LLMs with proxy rewards, including reward models (RLHF), rule‑based verifiers, and LLM‑as‑a‑judge (RLVR). The authors develop a theoretical link between parameter‑space flatness and proxy‑reward accuracy via a bound on excess Bradley–Terry loss, argue that “reference resets” implicitly enact GR, and introduce an efficient finite‑difference GR implementation for policy optimization. Empirically, across TL;DR summarization, AlpacaFarm RLHF, and GSM8K/MATH (rule‑based and LLM‑judge), GR improves robustness and final performance relative to common KL penalties, reference resets, or no regularization, and visibly mitigates judge/format hacking.

**Compliance With Llm Reviewing Policy:**

Affirmed.

**Key Questions For Authors:**

- Can you provide compute‑matched comparisons (steps, tokens, wall‑clock) between GR, KL, and reference resets, and report quality–compute trade‑offs? In particular, how does explicit GR compare to resets when training time is equalized?

- How sensitive are results to the exact KL implementation in GRPO (e.g., reverse‑KL in reward vs as loss, unbiased vs surrogate estimators)? Could you include a “principled” KL baseline per recent gradient‑equivalence analyses?

- Could you report multi‑seed means and confidence intervals for key tables/figures and, where possible, cross‑evaluator results (e.g., different LLM judges or human evaluation) to mitigate evaluator‑overfitting concerns on AlpacaFarm?

- Have you compared GR to SAM‑style updates or to recent stability‑focused algorithms like GVPO or TR‑GRPO on GSM8K/MATH? Such comparisons would help position GR among alternatives.

**Limitations:**

- **The “reference resets act like GR” claim is not fully proven**: The paper does not yet provide a strong formal derivation or a sufficiently controlled ablation to isolate whether resets really work because of implicit gradient regularization, rather than some other stabilization effect.

- **Experimental rigor is somewhat limited**: Many results appear to be single-seed, with limited confidence intervals or significance testing. Since RL post-training can be noisy and unstable, this weakens confidence in how robust the reported gains really are.

- **Missing comparisons to closely related methods**: The paper does not directly compare against several recent and relevant stabilization methods such as GVPO, TR-GRPO, SAM-style sharpness-aware optimization, or reward-refinement approaches like Clip/Delta.

**Strengths And Weaknesses:**

**Strengths**

1. The paper offers a clear and original framing: rather than directly constraining policy updates with an explicit KL penalty, it steers optimization toward regions where the proxy reward remains accurate by biasing learning toward flatter solutions via gradient regularization.

2. The paper includes useful diagnostic analyses. In particular, gradient norm is shown to correlate with sharpness and proxy-reward accuracy, and the onset of reward hacking appears to coincide with spikes in gradient norm.

3. The work addresses an important practical issue in current LLM post-training pipelines, where KL penalties are often expensive, tuned out, or insufficient, yet reward hacking remains prevalent.

**Weaknesses**

1. The claim that reference resets act as an implicit form of gradient regularization is interesting, but currently feels more empirical and interpretive than formally established. A stronger derivation or more targeted controlled ablations would strengthen this point.

2. The statistical rigor is somewhat limited. Many results appear to rely on single-seed runs, and confidence intervals or significance tests are largely absent, making it difficult to assess the robustness of the reported gains.

3. In the AlpacaFarm experiments, the same model family (GPT-4.1-Nano) appears to provide both training feedback and evaluation, which raises the possibility of evaluator overfitting or entanglement between training and assessment.

4. The discussion of recent nuances in KL implementation and their effect on stability and regularization in GRPO-style methods is limited. A more thorough treatment, including recent gradient-equivalence analyses, would help clarify how distinct the proposed method is from existing regularization effects.

---

> ### Author Rebuttal · Authors · 2026-03-30
>
> We would like to thank the reviewer for their helpful comments.
>
> >Q1: Compute-Quality Tradeoff
>
> Thank you for the recommendation, the Table below shows the winrate and runtime before early-stopping on the three different seeds evaluated for Qwen2.5 1.5B on the Alpaca dataset. We note that hyper-parameters were tuned for best winrate, not compute efficiency. GR converges in a shorter time to a better solution than a KL-penalty in all three seeds, and a better one than Resets on 2 of 3.
>
> | Time / Winrate | Seed #1           | Seed #2            | Seed #3
> |----------------|-------------------|--------------------|-------------------|
> | KL             | 29.9 mins / 25.1% | 41.1 mins / 27.6%  | 19.8 mins / 38.5%
> | Reset          | 32.9 mins / 22.3% | 35.4 mins / 27.1%  | **14.8 mins / 42.4%**
> | GR             |  **22.1 mins / 28.3%** | **8.5 mins   / 29.2%** | 17.3 mins / 40.2%
>
> >Q2: KL implementation sensitivity
>
> We use the standard KL implementation provided by TRL, which for GRPO is the “k3 in loss” estimator as used by SOTA labs e.g. [Deepseek](https://arxiv.org/abs/2501.12948) and [NVIDIA](https://docs.nvidia.com/nemo/rl/0.2.1/guides/grpo.html#on-policy-kl-approximation-use-on-policy-kl-approximation).
> [(Shah et al. 2025)](https://arxiv.org/abs/2512.21852) shows that k3 in loss is one of the two stable estimators, so we would consider it a principled baseline.
> We will specify the used implementation in the paper.
>
> > Q3: Number of random seeds
>
> We note that we replicated the Qwen 1.5B experiments three times, where GR performs better than the standard KL penalty in all seeds and better than resets on 2 of 3 seeds, as shown in the Table above and Figure 5 in the submission.
>
> We also now ran three additional seeds for each Qwen2.5 0.5B GSM8K experiments, in the rule-based and LLM judge settings, showing a consistent improvement across random seeds:
>
> Rule-based:
>
> | Method | Seed #1 | Seed #2 | Seed #3 | Avg + STD  |
> |--------|---------|---------|---------|------------|
> | No Reg | 51.9%   | 52.5%   | 52%     | 52.1%±0.3% |
> | KL     | 43.5%   | 45.6%   | 43.6%   | 44.2%±1.2% |
> | GR     | **56.7%**   | **55.0%**   | **56.8%**   | **56.2%±1.0%** |
> LLM-Judge:
>
> | Method | Seed #1 | Seed #2 | Seed #3 | Avg + STD  |
> |--------|---------|---------|---------|------------|
> | No Reg | 24.0%   | 16.5%   | 21.6%   | 20.7%±3.8% |
> | KL     | 23.4%   | 23.1%   | 26.1%   | 24.2%±1.7% |
> | GR     | **37.9%**   | **41.8%**   | **43.4%**   | **41.0%±2.8%** |
>
> > Q3: Same model family in both training and evaluation on AlpacaFarm
>
> Following [Gao et al (2022)](https://arxiv.org/abs/2210.10760), we do not use GPT4.1 Nano during RL training, we only use it as true reward $R^*$ to label pairwise comparisons for the reward model training data and for the final evaluation, analogously how we might use the same population of human labellers for data labeling and final evaluation.
>
>
> >Q4/L3: Have you compared [...] to recent stability‑focused algorithms like GVPO or TR‑GRPO on GSM8K/MATH? [and Clip-Delta]
>
> No, we do not believe these are relevant baselines in our setting.
> Clip-Delta [(Gao et al. 2024)](https://arxiv.org/abs/2410.15115) is a method specifically for process reward models (PRM). Our paper does not use PRMs, so clip-delta can not directly be applied to our problem setting.
> [GVPO](https://arxiv.org/abs/2504.19599) aims to stabilize training in off-policy settings, while we use a fully on-policy setting in our experiments.
> TR-GRPO and GVPO both focus on optimization stability in RLVR, while our method specifically targets reward hacking and is thus largely orthogonal.
>
> >Q4: Have you compared GR to SAM‑style updates
>
> GR is indeed very similar to SAM, even identical for specific parameterizations [(Karakida et al. 2023)](https://dl.acm.org/doi/10.5555/3618408.3619056). We thus assume that SAM would perform similarly to GR and have discussed this connection in the related work in Appendix A.
>
> > “reference resets act like GR” claim is not fully proven
>
> We agree with the reviewer, we will adjust the language in our draft to state this limitation more clearly.
>
> Thank you very much for the review, we believe our draft is stronger with the recommended revisions!

---

> > ### Author Rebuttal · Reviewer_G1in · 2026-04-04
> >
> > Thank you for the rebuttal. The added results and clarification of SAM, and different seed experiments resolved most of my concerns. Overall, the rebuttal resolves my main concerns, and I encourage the authors to reflect these clarifications in the revision.

---

> > > ### Author Response · Authors · 2026-04-05
> > >
> > > Thank you for the rebuttal acknowledgement, we are glad that we could resolve your concerns. We will update our submission accordingly.
> > >
> > > We would be grateful if you could increase your score from the current weak reject, or let us know what experiments we could run or remaining concerns we could address.

---

### Decision · Program_Chairs · 2026-04-30

**Decision:**

Accept (regular)

**Comment:**

This paper proposes gradient regularization as a way to prevent reward hacking. The review acknowledge the contributions although some results need to be strengthened. The authors should address the comments in the final version.